# Transient hydrodynamic effects influence organic carbon signatures in marine sediments

Clayton R. Magill[1,2], Blanca Ausín[1], Pascal Wenk[1], Cameron McIntyre [1,3,7], Luke Skinner[4], Alfredo Martínez-García [1,5], David A. Hodell[4], Gerald H. Haug[1,5], William Kenney [6] & Timothy I. Eglinton [1]

Ocean dynamics served an important role during past dramatic climate changes via impacts on deep-ocean carbon storage. Such changes are recorded in sedimentary proxies of hydrographic change on continental margins, which lie at the ocean–atmosphere–earth interface. However, interpretations of these records are challenging, given complex interplays among processes delivering particulate material to and from ocean margins. Here we report radiocarbon ($^{14}$C) signatures measured for organic carbon in differing grain-size sediment fractions and foraminifera in a sediment core retrieved from the southwest Iberian margin, spanning the last ~25,000 yr. Variable differences of 0–5000 yr in radiocarbon age are apparent between organic carbon in differing grain-sizes and foraminifera of the same sediment layer. The magnitude of $^{14}$C differences co-varies with key paleoceanographic indices (e.g., proximal bottom-current density gradients), which we interpret as evidence of Atlantic–Mediterranean seawater exchange influencing grain-size specific carbon accumulation and translocation. These findings underscore an important link between regional hydrodynamics and interpretations of down-core sedimentary proxies.

[1] Geological Institute, ETH Zürich, Zürich 8092, Switzerland. [2] Lyell Centre, Heriot-Watt University, Edinburgh EH14 4AS, United Kingdom. [3] Laboratory for Ion Beam Physics, ETH Zürich, Zürich 8093, Switzerland. [4] Department of Earth Sciences, University of Cambridge, Cambridge CB2 3EQ, United Kingdom. [5] Max Planck Institute for Chemistry, D-55128 Mainz, Germany. [6] Land Use and Environmental Change Institute, University of Florida, Gainesville, FL 32611, United States. [7] Present address: Scottish Universities Environmental Research Centre (SUERC), East Kilbride G750QF, United Kingdom. Correspondence and requests for materials should be addressed to C.R.M. (email: C.Magill@hw.ac.uk)

P revious studies suggest that dramatic climate changes during glacial terminations, including the last deglaciation, about 23–9 thousand years (kyr) ago, were related to periods of Atlantic meridional overturning circulation slowdown[1,2]. These slowdowns were accompanied by pronounced hydrographic changes[3] throughout the sub-tropical Atlantic, which are exemplified by paleoceanographic reconstructions off the southwest Iberian margin[1,4] (Fig. 1). Respective reconstructions demonstrate an antiphase relationship between North Atlantic Deep Water (NADW) formation and the strength of Mediterranean Outflow Water (MOW) during glacial termination[5,6], which together are important influences on Eurasian and high north-latitude climate conditions[7].

Microfossils and biogeochemical data suggest that marked paleoceanographic changes occurred in sub-tropical Atlantic circulation at intermediate water depths during glacial termination[2,8–12], but debate persists about the nature of these changes vis-à-vis apparent disparities among proxies within singular or correlative sediment horizons;[13] for instance, sea-surface temperatures (SSTs) reconstructed from co-deposited (organic) bio-markers and foraminifera[12,14,15]. There is some evidence suggesting particle re-suspension and lateral translocation (advection) of sediment[16] within nepheloid layers[10,17] drive apparent disparities among proxies with differing grain-size associations via sediment (hydrodynamic) sorting processes[18–20], but this evidence remains inconclusive as sediment mobilization and advection is often stochastic[16] and difficult to resolve[21].

Here, we reveal the effects of hydrodynamics on carbon accumulation in sediment from southwest Iberian margin core-sediments spanning the last ~25 kyr via complementary physico-chemical (e.g., grain-size distributions and X-ray fluorescence [XRF]) data and $^{14}$C ages measured for organic carbon (OC) in different grain-size classes and foraminifera separated from a 3.44 m-long sediment (Kasten) core recovered in 2013 during cruise JC089 aboard the RSS James Cook in the northeast Atlantic Ocean (Fig. 1). The core site called SHAK06-5K (37.571 °N, 10.153 °W, 2646 mbsl) lies at the lower slope of the southwest Iberian margin, where high sedimentation rates resolve paleoceanographic conditions over decadal-to-orbital timescales[4,9].

## Results and discussion

### Radiocarbon ages of bulk organic carbon and foraminifera.
Radiocarbon ages of OC increase with depth in every grain-size class and feature a maximum average radiocarbon age of 20,600 ± 900 yr, when coincident foraminifera reach their maximum calendar age of 21,725 yr BP (Fig. 2). Associated bulk sediment OC $^{14}$C feature a consistent down-core difference against foraminifera radiocarbon ages of 1450 ± 200 yr (Fig. 2) with one exception at 65–66 cm (~600 yr offset [Supplementary Data 1]) that fell during prominent Mediterranean sapropel 1 formation[22] wherein radiocarbon age offsets appear influenced by particularly 'old' foraminifera calendar ages as opposed to unexpectedly young gross sediment TOC. In contrast, the radiocarbon offset between organic carbon in clay-size sediment fractions against foraminifera ($R_{C–F}$) is more variable, with down-core differences of between about 0–2000 yr (Fig. 3a). The radiocarbon offset between organic carbon in coincident fine or coarse-silts and foraminifera ($R_{FS–F}$ and $R_{CS–F}$, respectively) likewise is variable, with down-core differences of ~1000–3500 yr (Fig. 3a). Because organic carbon in sediment fractions of >63 μm is subject to varied influences of biomineral-bound OC (ref. [23]) and incomplete disaggregation of flocculates with effective diameters in excess of 100 μm (refs. [19,24,25]), we will focus here on changes in down-core radiocarbon offsets apparent between coincident finer sediment fractions and foraminifera. Notwithstanding questions

about the reliability of differently preserved tests (translucent versus frosty), prior studies suggest foraminiferal $^{14}$C dates are an accurate indication of initial deposition age in rapidly accumulating sediments because of their consistent high settling velocities, whereas co-deposited finer sediments are more subject to (re)suspension with attendant spatio-temporal biases on sedimentary proxies[20,26].

The organic carbon in all finer sediment fractions (i.e., C, FS and CS) has older radiocarbon ages as compared to coincident foraminifera, and the magnitude of their respective radiocarbon offsets change in step with complementary proxies of (paleo) oceanographic variability since at least 25 kyr ago (Fig. 3; Supplementary Data 1). A lower radiocarbon offset (i.e., decreased $R_{C–F}$, $R_{FS–F}$ and $R_{CS–F}$ values) during the Last Glacial Maximum (LGM; 23.2 kyr ago) transitions into more moderate offsets of ~2000 ± 500 yr during preliminary glacial termination, and then rapidly peak to values of ~1500–3500 yr amid the middle of Heinrich Event 1 (HE1; 17.5–14.7 kyr ago). After an interim of lower radiocarbon offset through the conclusion of HE1 and initial Bølling/Allerød interstadial (B/A; 14.7–12.8 kyr ago), the relative age differences among grain-size classes increase again at B/A-to-Younger Dryas (YD; 12.9–11.7 kyr ago) transition with moderate offsets of ~2000 ± 500 yr. Low-to-intermediate average radiocarbon offsets subsequently persist through the mid-Holocene. Thereafter, respective offsets climb to significantly greater values amid the last few millennia, most likely due to anthropogenic impacts on sedimentary processes[27] (e.g., particulate material transmission and deposition)[28].

### Potential driver(s) of $^{14}$C differences.
Previous studies indicate systematic radiocarbon offsets among grain-size classes could be consequent to several different factors[29], including preferential bioturbation[30,31], diagenetic alteration or downslope mobilization of foraminiferal tests[8], and the differential lateral transfer of bottom and intermediate nepheloid layer sediment fractions[32] via deeper water currents[16,20,33]. However, although benthic organisms can induce age or size-dependent depositional displacement[30,31], bioturbation effects are unlikely to affect down-core records significantly, given ichnofabric evidence of low-moderate degree[34] of mid-tier[35,36] (limited to upper <10 cm of the substrate) bioturbation alongside relatively high sedimentation rates[31,37] and lack of a deep mixed layer[29] as indicated by $^{210}$Pb from multi-cores at SHAK06-5K (Supplementary Data 3). This evidence is supported by geochemical biodiffusion models[38,39], which further suggest limited biodiffusive coefficients of ~0.15 ± 0.05 cm$^2$ yr$^{–1}$ at SHAK06-5K (Supplementary Fig. 4). Diagenetic alteration or downslope remobilization of foraminiferal tests are also unlikely, given coincident foraminifera are consistently younger versus organic carbon and that tests are resistant to winnowing[19].

There is a moderate correlation apparent between down-core values of $R_{FS–F}$ and $R_{CS–F}$ and XRF-derived Zr/Al ratios of bulk sediment (Fig. 3d–f), which can often serve as a proxy of MOW flow-core velocities[40], that implies bottom-current flow dynamics are an important factor in controlling grain-size specific re-suspension. Yet, the recent MOW flow-core shows an average depth of ~500–1500 mbsl (ref. [41]) that lies well above SHAK06-5K (2646 mbsl). Thus, while at least the base of the MOW descended to at least 2600 mbsl (Fig. 1) during past Heinrich events[5,41], it is improbable that Zr/Al trends at SHAK06-5K directly proxy past changes in past MOW flow-core velocities[13]. Therefore, we compared the differences in normalized[4,40,42] Zr/Al ratios between marine cores recovered from a site close to the Strait of Gibraltar (U1389; 36.425 °N, 7.277 °W, 644 mbsl), which is a sensitive recorder of MOW flow velocities[40,42], and

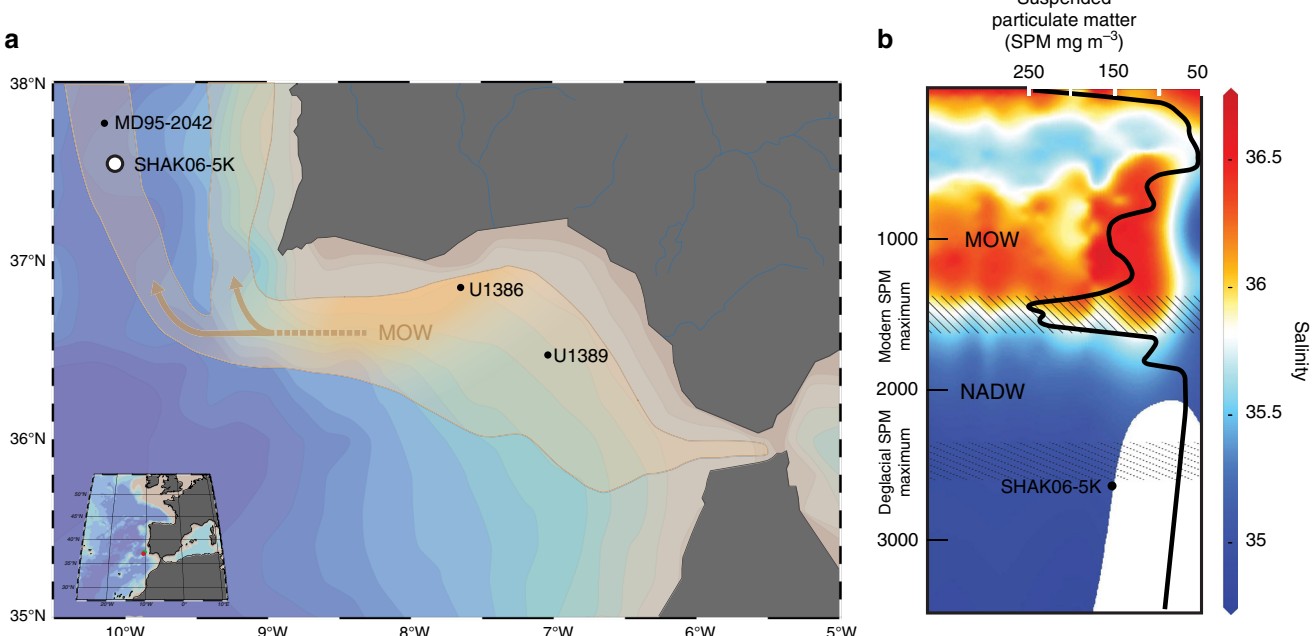

**Fig. 1** Core sites and down-column water chemistry profiles. Maps of the southwest Iberian margin (**a**) showing modern ocean current trajectories of Mediterranean outflow water (MOW [shown in orange]), which flows around 500–2000 m depth, and locations of SHAK06-5K (open circle) and MD95-2042, U1386, and U1389 (black dots). Orange arrows show approximate sediment transfer courses out of the Gulf of Cadiz[9,61,65,84]. Bathymetric contours are shown at 50 m intervals as adapted from Ocean Data View (Schlitzer, R. Ocean Data View, odv.awi.de, 2018). **b**, Contemporary profile around 37.5 °N for salinity, which correlates with seawater density[50]. Also shown are estimates of the modern (solid hatches) and deglacial (stipled hashes) nepheloid layer mixing (peak) depths[16,17,50,84], which are reflected by particulate-matter concentration maxima. Deglacial nepheloid layer depths are derived from numeric simulations[50,84] considered together with complementary grain-size, isotopic and foraminiferal data regarding glacial MOW dynamics[3,8,50]. NADW, North Atlantic Deep Water

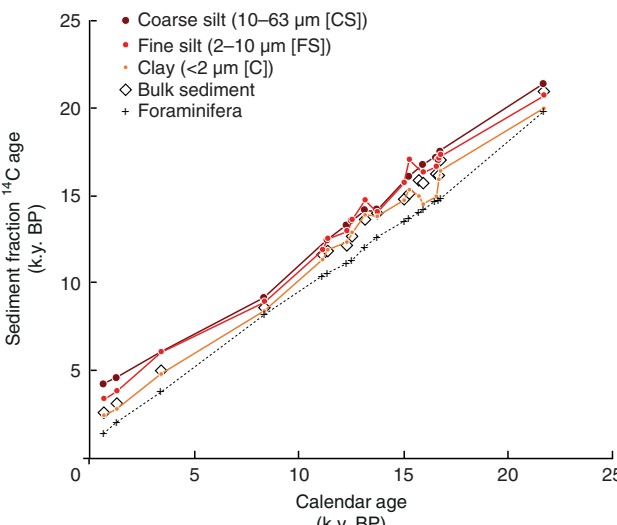

**Fig. 2** Down-core age relationships among grain-size sediment fractions. Down-core relationships in conventional $^{14}$C ($R$) age of organic carbon in bulk sediment and among grain-size classes as a function of calendar age derived from $^{14}$C measurements on coincident planktonic foraminifera (c.f., Supplementary Data 1). Between sediment-fraction differences are magnified in Fig. 3 and furthermore available in Supplementary Data 1

another site closer to the moat generated by the uppermost MOW (U1386; 36.828 °N, 7.755 °W, 561 mbsl), which is sensitive to both MOW flow velocities and flow-core depth[42]. Considered together, such differences should be a robust, though indirect, indicator of MOW flow-core depths[42,43]. Not too surprisingly,

patterns of U1389–U1386 differences show a stronger correlation with $R_{FS–C}$ and $R_{CS–C}$ trends as compared to either core alone (Fig. 3f). With this in mind, here we suggest that Zr/Al instead tracks changes in advected finer-grained terrigenous siliclastics entrained in association with a nepheloid layer[16,17,44,45].

There are significant low-to-moderate strength relationships apparent between radiocarbon offsets and XRF (i.e., manganese-to-aluminum ratio [Mn/Al]) trends at SHAK06-5K (Fig. 3e; Supplementary Data 1) that hint toward the effects of bottom water oxygenation with respect to sediment flux and OM degradation. Previous studies of Mediterranean seawater and sediment dynamics suggest Mn/Al trends in regional deep-sea sediments correlate with redox conditions[22] that, in turn, impact the abundance and degradation (e.g., 'pre-aged') of OM in sediments and suspended particulate matter (SPM) through cyclic oxygen (re)exposure[46,47]. More specifically, paleoceanographic reconstructions of deglacial MOW fluctuations suggest that there was increased discharge of deep-and-intermediate waters[48] with high amounts of fine sediment fractions and TOC[22,49].

Insomuch as nepheloid layer dynamics and bottom current flow velocities along the Iberian margin are each related to seawater density gradients[44] between regional MOW and Atlantic seawater[41,50], increased Zr/Al and Mn/Al ratios most likely parallel the enhanced lateral transport flux and deposition of finer sediments (i.e., silts) as compared to vertical input of fresh hemipelagic materials[22,48] because of the deeper, enhanced nepheloid layer developed between seawater masses with disparate densities[16,17]. This interpretation is consistent with the nominal differences in down-core sortable silt distributions (Supplementary Data 1) and is reinforced by parallel down-core records of the unsupported $^{231}$Pa-to-$^{230}$Th ratios at SU81-18

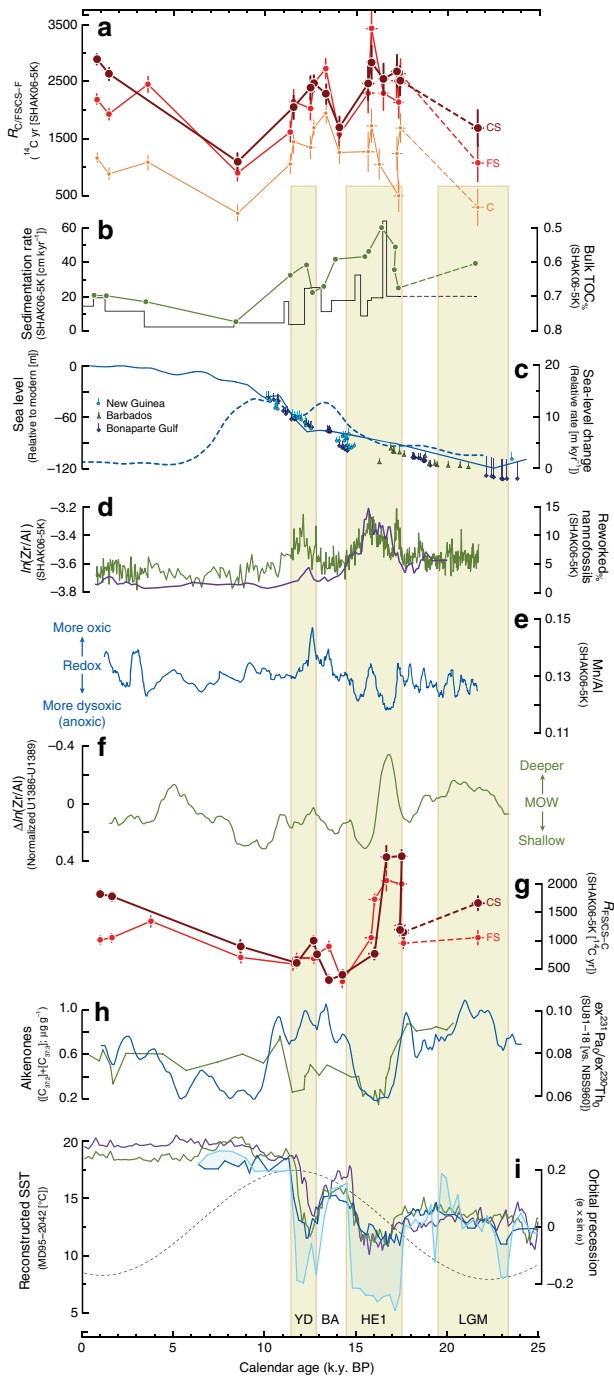

**Fig. 3** Deglacial trends in down-core records at SHAK06-5K and vicinal Iberian Margin ocean cores. Comparison of down-core sedimentary proxies at Iberian margin sites during glacial termination and Holocene. **a** Radiocarbon age offset between grain-size sediment fractions (clay [C], fine silt [FS], coarse silt [CS], foraminifera [F]; $R_{x-y} = R_x - R_y$, where $x$ and $y$ represent discrete sediment fractions [c.f., eq. 1]) isolated from the same sediment core intervals of SHAK06-5K. Also shown are propagated $1\sigma$ (s. d.) uncertainties of differences in radiocarbon age among grain-size sediment fractions (c.f., Supplementary Data 1). **b** Average sedimentation rates of the sediments at SHAK06-5K (black line [c.f., Supplementary Data 1]), together with bulk percent total organic carbon (TOC%) for respective sediments (N.B., the axis reversal for improved down-core comparisons). **c** Relative sea-level (RSL) fluctuations of the southeast Portuguese margin[27,28] (blue solid line) beside corresponding global RSL data with depth uncertainties[85]. The rate of sea-level change is also shown, as modeled in Monte Carlo experiments with 6 m coral depth uncertainty[86]. **d** Zirconium-to-Aluminum (Zr/Al) ratios of bulk sediments at SHAK06-5K (green solid line). Raw values are natural-log transformed to improve data normality. Also shown are relative abundance (percentage) of reworked nannofossils in down-core sediments at SHAK06-5K (c.f., Supplementary Data 4). **e** Manganese-to Aluminum (Mn/Al) ratios of bulk sediments at SHAK06-5K, which serve as a relative indicator of benthic redox conditions[22]. **f** Difference between normalized Zr/Al ratios at U1386 and U1389 (ref. 40) that theoretically presents a surrogate measure of MOW flow depth[43]. **g** Radiocarbon age offset between co-occurring grain-size classes as compared to clays at SHAK06-5K. Again, corresponding propagated $1\sigma$ uncertainties of differences in radiocarbon age among grain-size sediment fractions are also shown (c.f., Supplementary Data 1). **g** Alkenone concentrations ($C_{37:2} + C_{37:3}$ [light blue solid line]) for at MD95-2042 (ref. 14); higher values indicate increasing primary production or increases in organic matter preservation[55]. Also shown is excess $^{231}$Pa-to-$^{230}$Th ratio at SU18-81 (green solid line), demonstrative of bottom-water circulation strength[2]. **h** Parallel down-core reconstructions of sea-surface temperature at MD95-2042 derived from coincident foraminifera assemblages[67] (purple solid line), alkenones (light blue[67] and dark blue[14] solid lines, respectively) and tetraethers[14] (green solid line). Also shown is calculated orbital precession (dashed black line), which equals the product of calculated eccentricity ($e$) and the sine function of longitude of the perihelion ($\omega$). Dashed lines connecting points represent intervals of low data resolution and high meltwater release, which could drive anomalous radiocarbon values[1,83]. BA Bølling/Allerød, HE1 Heinrich Event 1, LGM Last Glacial Maximum, YD Younger Dryas

(37.767 °N, 10.183 °W, 3135 mbsl) (Fig. 3h). Thus, we suggest that it is not MOW flow-core velocity, in itself, that drives apparent Zr/Al trends at SHAK06-5K, but the associated nepheloid layer(s) that develops alongside wider salinity gradients amid Atlantic-Mediterranean seawater masses[17,44] at lower depths[50]. This association squares with observations of turbulent (re)suspension of finer sediment from upper slopes at their contact with uppermost MOW flow (c.f., internal tides)[16,51] that then settles at MOW–NADW interface[16,17,22,52] along with SPM transported from up-current distal locations[9,16,22,48]. Aforementioned drivers also square with the moderate correlation strength between radiocarbon offsets and down-core sedimentation rate (Fig. 3a, b) and inferred benthic redox conditions (Fig. 3e), both of which are entwined to changes in sea-level[22]. This combination

of (hydro)physical drivers would drive increased injection of turbid Mediterranean water masses into the Gulf of Cadiz at high velocities during periods of deglacial transgression[6,48,49], alongside regional increases in rainfall (i.e. river discharge) and benthic dysoxia[22,48,49]. As such, times of stronger MOW flow would thus result in more advection of fine grained terrigenous clastic material to our site, which then would be admixed together with local hemipelagic rain in varying proportions[18,49,53].

Although the exact balance of MOW temperature-salinity parameters during glacial termination remains uncertain[41,54], reconstructed densities ($\sigma$) of important Atlantic-Mediterranean seawater masses are more certain. The contrast between reconstructed densities of MOW and NADW through Greenland stadials ($\sigma_{MOW} - \sigma_{NADW}$ = 2.4 kg m$^{-3}$) is much larger as compared to warmer (interstadial) intervals[41], such as the recent Holocene[45,54] ($\sigma_{MOW} - \sigma_{NADW}$ = 0.7 kg m$^{-3}$). Considered together with evidence of low-moderate degree[34] of mid-tier (< 10 cm of the substrate) bioturbation[35,36], our data suggest that down-core radiocarbon age offsets among grain-size classes are consequent – at least in part – to differential hydrodynamic

effects on sediment lateral transfer vis-à-vis differences in regional Atlantic-Mediterranean seawater densities and flow depths. All data considered together, we suggest there are competing influences of (hemi)pelagic dilution vs. advection of finer sediment fractions as a function of MOW depth and OM degradation extent that, in turn, are related to regional (Mediterranean) terrestrial hydroclimate, ocean circulation dynamics, and benthic oxygenation.

**(Paleo)oceanographic implications**. Considerable changes occurred in regional MOW dynamics (i.e., current depth and flow velocities) during glacial termination[9,40,42,43], about 20–10 kyr ago; however, important details of these changes remain unclear. Increasing grain-sizes[42] and increases in reworked nannofossil influx[11,55] (Fig. 3d) during deglacial transitions and the mid-Holocene correspond to intervals of decreased Nile river discharge[56], which led to more saline (denser) MOW[41,45,50], and decreased formation of less dense, cold NADW[54,57]. Intervals associated with cold Arctic conditions[2,54,56] show decreased percent total OC (TOC$_\%$ [Fig. 3b]) together with overall high sedimentation rates (Supplementary Data 1), which typically parallel OC burial efficiency[37]. Although relative differences in terrestrial input (vs. marine) could be one explanation of these observations, it does not befit the consistent low ratio values of bulk carbon-to-nitrogen (Supplementary Data 1) and branched isoprenoid tetraethers[14] (BIT) that together indicate negligible soil OM input throughout the last ~25 kyr. Decreased TOC$_\%$ through these intervals also does not track the significant, though varied, disparities in relative abundances or reconstructed temperatures among principle oceanography proxies[14] (i.e., alkenones, isoprenoid tetraethers, and foraminifera [Fig. 3h, i]).

One approach to reconcile such apparent discrepancies invokes recent data suggesting grain-sizes of 10–63 μm (c.f., CS) feature especially protracted lateral transport histories in ocean margin systems[33,58,59], such that CS fractions contain decreasing proportions of fresh organic matter and decreased TOC$_\%$ through time via progressive pre-depositional degradation[23,33]. During progressive oxic degradation, residual OC in CS fractions will become increasingly pre-aged because of decreased fresh organic matter and its accumulative residence in nepheloid layers over repeated (re)suspension–deposition cycles[21,33,58–60].

Interestingly, previous studies reveal that the recent MOW transfers particulate matter with an average diameter of 5–25 μm (refs. [17,61]) from coastal Atlantic sources (Gulf of Cádiz)[55,61] that then is entrained to more distal locations of the continental Iberian margin. Assuming grain-sizes of 2–10 μm (c.f., FS) have similar sediment transport histories as coarser silt[51,59], these combined data suggest that $R_{FS–C}$ and $R_{CS–C}$ trends (Fig. 3g) are related to relative differences in organic carbon mixing proportions and degradation among grain-size classes as controlled by pre-depositional translocation (entrainment) time, although the specific mechanisms responsible for apparent bulk $^{14}$C differences remain speculative in lieu of biomarker compositional data in corresponding grain-size sediment fractions.

Our analyses reveal down-core coherence of TOC$_\%$ and the corresponding proportion of OC derived from clay ($r = 0.789$ [Supplementary Data 1]) that drive strong parallels of TOC$_\%$ against $R_{CS–C}$ and $R_{FS–C}$ (Fig. 3b, g). In contrast, TOC$_\%$ has weak correlations with both fine- and the coarse silt fractions (Supplementary Data 1). As such, marked decreases of TOC$_\%$ during parts of HE1 and YD (Fig. 3b) follow alongside a relative shift towards more refractory, pre-aged OC in coarse and fine-silt fractions (Fig. 3g, Supplementary Data 1) independent of OC dilution or isotopic mass balance. Assuming particle entrainment during deglacial formation of MOW-related nepheloid layers was

also dominated by grain-sizes of 15 ± 10 μm (ref. [17]), these data suggest there was increased lateral (re)suspension of finer sediment[19,52,59,62] (i.e., silt) with relatively pre-aged OC[21,33,58] derived from more remote allochthonous sources[63] such as marginal Gulf of Cadiz drift deposits[61,64,65].

**Apparent discrepancies among (paleo)oceanography proxies**. Differential lateral transfer dynamics among grain-size classes may help to explain apparent disparities among proxies during paleoceanographic reconstructions in drift deposits when a single age-depth model is adopted for all down-core records (c.f., Fig. 3i). For example, earlier studies reveal disparities in sea-surface temperature (SST) estimates reconstructed from alkenones[14,55], glycerol dialkyl glycerol tetraethers[14,66] (GDGTs), and foraminifera[12,15,67] from correlative down-core records at MD95-2042 (ref. [14]). Although some of these disparities could be consequent to multiple or independent factors[14,66] (e.g., phylogenetic or other species-dependent factors), degradation and hydrodynamic effects could have an intrinsic role in explaining such proxy paradoxes[13,59,63,68].

Overall MOW flow velocities decline from >250 cm s$^{-1}$ at the strait of Gibraltar to about 10–15 cm s$^{-1}$ off Cape St. Vicente spur and <10 cm s$^{-1}$ along the western Portuguese slope[65]. Considered together with observations that demonstrate resuspension of fine silts and benthic aggregates occurs when respective currents exceed 15-to-25 cm s$^{-1}$ (refs. [19,59,62]), this suggests a critical threshold for deceleration is crossed in the vicinity of SHAK06-5K that might lead to differential degradation[24,29,33,58] and deposition of advected SPM from up-current locations[65]. Accepting previous studies showing prototypic organic matter aggregates and fine surface-sediments of the southwest Portuguese margin show average settling speeds of ~0.015 cm s$^{-1}$ within associated MOW branches and spend ~50% of time in (re)suspension[52,59], these combined data insinuate advective displacement distances of up to several hundreds of kilometres (c.f., Alboran Sea[25]) that befit the results of bottom boundary (BOBO) landers[53] and bottom-current circulation simulations[65].

Alkenones and GDGTs show dissimilar radiocarbon age offsets[26,69,70] as compared to coincident foraminifera in north Atlantic drift deposits[20] that befit the combination of their differing grain-size associations[31,59] and degradation recalcitrance[29,68,70]. Previous studies indicate alkenones, which are more recalcitrant to oxidation[20,26,70], occur in association with sedimentary particles of <6 μm (refs. [71,72]), but less-recalcitrant GDGTs[66,69] are associated with sedimentary particles of 6–32 μm (ref. [73]). Although sorting processes are subject to variable and differential influences among grain-sizes (e.g., particle sphericity, particle aggregation, and turbidity [particle concentration])[59,60] and between sedimentary particles of the same size (e.g., grain mineralogy [illite vs. kaolinite])[74], analogous molecular sediment-fraction associations[23,29] – and thus relative age offsets[25,33,45,64] – are also implied for southwest Portuguese margin sites[71], wherein down-core alkenone and GDGT-derived SST trends are offset (lagged), like $R_{CS–FS}$ values, about 200 yr on average, and likewise are offset from coincident foraminifera-derived SSTs up to several hundreds of years (Fig. 3g, i).

The magnitude of time offset between organic sedimentary proxies of SST through deglaciation is consistent with the average radiocarbon age offsets of their corresponding grain-size classes (i.e., alkenones [FS] and GDGTs [CS]) (Fig. 3a, i). Stepwise increases in instantaneous (differential) lag phase amid HE1 and YD happen in concert with high radiocarbon offsets (Supplementary Fig. 3), high MOW flow velocities and densities (Fig. 3d–f), and elevated fluxes of reworked nannofossils[11,55]

(Fig. 3d) despite rather uniform[4,45] corresponding grain-size distributions at SHAK06-5K (Supplementary Data 1). Like some other studies[13,59], these same increases do not feature changes in median diameter of the sortable silts (equivalent to coarse silt [CS] fraction), which are often used to reconstruct flow velocities[9]. Further work, especially paired [14]C measurements of differing proxies (e.g., alkenones) and their carrier phase (e.g., coccolithophores), is essential to establish the occurrence and significance of these phenomena. Even so, our study provides support for significant hydrodynamic effects on organic carbon transport, degradation, and deposition on ocean margins, and interpretations of related (paleo)climate records.

To summarize, organic radiocarbon age differences among grain-size classes as compared to coincident foraminiferal tests in marine sediments of the northeast Atlantic margin reveal differential lateral transfer dynamics accompanying particle mobilization, as controlled by paleo-current densities vis-à-vis nepheloid layer dynamics. Intervals with intensified Mediterranean Outflow, which closely parallel increased Atlantic-Mediterranean seawater density contrasts, amid Heinrich Event 1 have much higher radiocarbon offsets among grain-size classes of ~1000–2500 yr and lower organic carbon concentrations as compared to intervals with more sluggish Mediterranean Outflow amid the mid-Holocene. In consequence, our results suggest differential lateral transfer dynamics can influence apparent lead–lag patterns among proxies with differing grain-size associations; as such, hydrodynamic influences on organic carbon accumulation and transfer are important factors to consider in interpretations of diverse co-occurring proxies in down-core records, which can experience differential degradation and hydrodynamic (sorting) processes.

## Methods

**Sediment sampling procedures and fraction separation.** The entire core was sectioned at 1-cm resolution on-board, from which 21 discrete sediment intervals (~50 g) were sub-sampled, before storage at −20 °C. Sub-samples were separated through wetted fine-mesh sieves[75] and tube settling protocol[76] to create a series of four grain-size sediment fractions: clay (<2 μm [C]), fine silt (2–10 μm [FS]), coarse silt (10–63 μm [CS]), and sand (>63 μm [S]). Although sediment-fraction recoveries were not monitored directly, previous studies demonstrate wet-sieve recovery percentages exceed 85% both for mass and bulk organic carbon in most instances[60] and have a nominal influence on associated bulk sediment-fraction isotopic signatures[71,77] despite small, though significant, losses of dissolved organic matter during rinsing. Likewise, measured total bulk organic compositions correlate with mass-balance calculations of the organic composition among grain-size sediment fractions ($r^2 = 0.718$ [c.f., Supplementary Data 1 and Supplementary Fig. 2]). Well-preserved tests of the planktonic foraminifer *Globigerina bulloides*—abbreviated "F" for foraminifera—were subsequently picked from associated 200–250 μm sediment fractions.

**Sample analysis.** Radiocarbon measurements of foraminiferal tests and decarbonated (hydrochloric acid fumigated)[78] bulk sediment fractions were made on a mini-carbon dating system[79] (MICADAS) following graphitization or via elemental analyser as detailed previously[80]. Radioactive carbon isotope compositions are shown as conventional [14]C ages ($R \pm 1\sigma$ s.d.) to calculate relative age relationships (i.e., offsets)[81] among grain-size classes, where $x$ and $y$ represent discrete sediment fractions:

$$R_{x-y} = R_x - R_y \qquad (1)$$

To calculate absolute age relationships[26,33], however, conventional [14]C ages of foraminiferal tests were converted into calendar years in an age-depth model (CALIB 7.1)[82] with a dynamic marine inorganic carbon reservoir correction (Supplementary Data 1) that is consistent with chronostratigraphic constraints imposed by planktonic oxygen isotope records[1,2,83] from MD95-2042/MD99-2334K (37.799 °N, 10.168 °W, 3146 mbsl) to within several hundred years[83]. XRF (Avaatech [University of Cambridge]) analyses were used for semi-quantitative analysis at 0.5 cm depth intervals on u-channel (sub)cores extracted from composite split-core scans of SHAK06-5K Kasten core material (c.f., Supplementary Data 2).

**Age models.** Foraminiferal (*Globigerina bulloides*) test [14]C ages were used to construct the age model for SHAK06-5K (37.571 °N, 10.153 °W, 2646 mbsl). To do so, surface reservoir ages estimated from at MD99-2334K (37.799 °N, 10.168 °W, 3146 mbsl)[1] were subtracted from each conventional [14]C date (c.f., Supplementary Data 1) before conversion to calendar ages with CALIB 7.1 (ref. [82]). Then, respective calendar ages were used to construct down-core sediment depth-age models after stratigraphic alignment (c.f., Supplementary Fig. 1) against U1385 (37.571 °N, 10.126 °W, 2578 mbsl)[4]. With this in mind, the corresponding propagated 1σ uncertainties used for estimating phase relationships fall below about 150 yr with few exceptions (c.f., Supplementary Data 1). We note that these uncertainties do not have much influence on our interpretations of lead/lag phase among proxy records (c.f., Fig. 3g, i) because appertaining proxies (i.e, $R_{CFS/CS-C}$ and tetraether/alkenone-derived SST) are derived from singular cores (SHAK06-5K and MD95-2042, respectively) and thus are internally consistent.

## Data availability

The authors declare all the new data used to support this research are available within the article and its supplementary information files.

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

## Acknowledgements

We wish to thank Daniel Montluçon and Negar Haghipour for assistance with organic radiocarbon analyses, and Simon Crowhurst for down-core sediment XRF data. This work was made possible by a Marie Curie Actions postdoctoral fellowship to C.R.M and NERC support for cruise JC089 (NE/J00653X/1) to D.A.H. and L.S.

## Author contributions

C.R.M. and T.I.E. conceptualized the project. L.S., D.A.H. and T.I.E. were among the principal investigators on cruise JC089. C.R.M., B.A., A.M.-G., G.H.H. and T.I.E. supervised and interpreted the radiocarbon analyses conducted by P.W. and C.M. D.A.H. and W.K. supervised X-ray fluorescence and lead-210 analyses, respectively. C.R.M., P.W., and B.A. wrote the manuscript with substantial contributions from C.M., L.S., D.A.H. and T.I.E. All authors reviewed the manuscript before submission.

## Additional information

**Competing interests:** The authors declare no competing interests.

