## [Peer Review File · Nature Communications]

Reviewers' comments:

Reviewer #1 (Remarks to the Author):

The authors present a new data set on ^{14}C ages of bulk organic carbon (OC) in different grain size classes of sediments from the Iberian continental margin, which are compared with ^{14}C ages of planktic foraminifera taken to reflect depositional age. They observe large and variable offsets between the OC in different grain size fraction and the depositional age and suggest sedimentology to control these, which has further implications for expected leads and lags of proxy records associated with different grain size classes. This is a very interesting study taking a novel approach, which has great potential to improve the understanding of the effects of sedimentology on proxy records.

While I agree with the general hypothesis that differences in age of different grain size fractions likely reflect differential transport histories of the respective materials in response to their hydrodynamic properties, I am not convinced by the suggested mechanisms put forward by the authors to explain their observations.

First of all, I do not see the "strong positive relationship" (line 122) described for down-core values of age offsets between fine silt and foraminifera with XRF derived $\ln(\text{Zr}/\text{Al})$ values (as a side note: in the text often referred to as the linear ratio, while in the plots the logarithmic version is presented – this should be adjusted). While there seems to be an agreement for the broad HS1 interval (with some differences in timing, which might be explained by differences in the resolution of the two data sets), I do not observe such a relationship for the YD interval. Instead, there is a strong maximum in the age offset in the BA (or a local minimum defined by just one measurement around 14.5 kyr) followed by a decline during the YD toward the early Holocene. The authors should be more careful describing such apparent agreements, in particular since their sediment core has a very high sedimentation rate and, likely, a very good age control, as it is tuned to the very well dated record MD99-2334K (another side note: a better description of the age model used and its associated uncertainties should be provided, at least in the supplement; the figures would benefit by indicating age tie points for the records of this new core SHAK06-05K).

Lateral transport of the silt fractions (coarse and fine silt) is suggested to occur over long timescales (and distances) in the intermediate nepheloid layers, and current flow strength controls the amount of laterally supplied silt. This suggestion includes the inherent assumption that material transported in INLs is pre-aged. This requires INL material to be re-suspended after having been deposited elsewhere. Such re-suspended material is typically not transported very far, as these particles settle rather rapidly. I would like the authors to present evidence that pre-aged material is entrained in these INLs. Current velocities of the currents that are supposed to transport the silt fractions should also be considered in order to evaluate the likelihood of the proposed mechanism.

The sediment at this core site accumulated at high but variable rates. The sedimentation rate estimates are presented in the supplement. Rough inspection reveals that the variability of the sedimentation rates resembles strongly the records of age offsets presented in Figure 3A, suggesting a coupling of sedimentation rate and the processes responsible for the age offset. Moreover, this agreement is likely better than that with $\ln(\text{Zr}/\text{Al})$ ratios, which is discussed at length in the manuscript. The implications of this observation should be discussed.

As the core site is located at the continental margin, and the time period investigated includes a transition from low to high sea-level stand, the effect of shelf erosion induced by sea-level rise in re-distributing pre-aged sediment should be considered, too. Interestingly, the increase in age offsets (and the strongest maximum in age offsets between silt and clay) occurs approximately coevally with the onset of global sea-level rise at ~ 16.8 kyr as reconstructed by Lambeck et al., (2014). The authors should evaluate when exactly the (outer) shelf was flooded in the study area (using shelf bathymetry and the sea-level curve).

The data plots are presented without uncertainties, which may be intentional to ensure readability. However, at least for the relatively low resolution ^{14}C data, error bars should be displayed (Fig. 3A, F).

Overall, I think the study has a great potential to reveal the effect of sediment redistribution on proxy records, but more care must be taken in evaluating the possible mechanisms responsible for sediment transport, which each have a different implication.

Below I list a number of more specific comments:

Line 8: I think this should read "...delivering particulate material to continental slopes (or: ocean margins)..." rather than "ocean shelves", which are not considered in this study using a core retrieved from ~2.5 km depth

Line 10: "radiocarbon ...signatures measured for organic carbon in differing grain-size sediment fractions..."

Lines 53 and following: Please provide more detail on the grain size separation methods: Did you dry sieve or wet sieve? How did you monitor recovery? Are weighted ages of OC in the grain size classes equivalent to bulk OC ages? If not, what are the potential loss mechanisms, could they be differential between compounds of different age/between different size classes?

Line 74: Here, more details on the age model should be provided. Reword "absolute age relationships" to "The age model", as it is not clear what the "relationship" refers to (of what with what?).

Line 90: I suggest to move the word "here" to after "focus"

Line 106: here and elsewhere (e.g., lines 122, 138): the increase in age offset, according to Figure 3, occurs prior to the YD in the BA warm period.

Line 138: unclear what the terms RFS-y and RCS-y refer to.

Line 139: is there evidence that these fine-grained siliciclastics are indeed entrained in INLs off the Iberian margin? How far can they be transported?

Line 148: here the authors refer to a ²³⁰Th-normalized flux record, which is not presented in the figure in spite of a reference to Figure 3G.

Line 160-162: as explained above, I don't think that it is convincingly shown that transport occurs in INLs.

Line 169 and 202: The term "pleniglacial" is uncommon in paleoceanography. It is more often used in terrestrial paleoenvironmental studies of the Eurasian continent, and it refers to the full glacial period prior to the LGM, which is not well resolved in the records presented here. I suggest to use more common terminology and suspect that indeed the authors meant "deglacial"

Line 172: "declining percent total OC" is a bit misleading; "low" or "decreased" might be clearer.

Line 187: Is it possible to assess/estimate how long the silt particles would reside in the INLs? Is there information on the current velocities in the respective depth?

Line 195-196: Here the authors mention a relationship between TOC% and OC derived from clay; it would be nice to see a figure illustrating this relationship.

Line 196 and 199: Again, the "strong parallels" between the two records are not evident from the figure; likely the authors refer mainly to the early phase of the HS1, where the age offset shows a pronounced maximum paralleled by a strong minimum in TOC%. In the YD, there is no clear correspondence; while the TOC% record has a local maximum in the BA and a minimum in the early YD which slowly increases toward the early Holocene, the age offsets are slightly higher in the BA or early YD and slowly increase through the Holocene.

Line 206 and following: I am confused by this paragraph, and it is not clear to me what the authors conclude from it.

Line 217: Is the core site really a drift deposit? Is there sedimentological evidence for this?

Line 229: Does the grain size association of the biomarkers suggest more long-range transport of alkenones than of GDGTs?

Line 236: It would be helpful to specifically state which grain size class corresponds to which SST proxy.

Line 237 and following: If transport causes the lags in the biomarker SST records relative to the foram-derived one, a different origin/source area for the biomarkers during the periods of stronger lateral transport would be implied, which would likely result in a shift in SST estimates (unless it is very local).

Line 240: do you mean "in spite of rather uniform nominal grain-size distributions"?

Line 255: The YD does not show higher age offsets than the BA period! Rather, the opposite is the

case.

Figure 1: In panel B, the estimated depth of deglacial INL is marked. It would be useful to briefly explain what this reconstruction is based on.

Figure 2: The blue shading between the lines indicating the ages of the different grain size fractions does not help much and should be removed. The age differences are better visible in Figure 3. Axis label "Sediment Fraction 14C Age" should use the unit "ky BP", and axis label "Calendar Age" should be in "ky" (an "age" cannot be "ago")

Figure 3: In panels A, B, and F, the errors associated with the measurements should be displayed. In the caption, the reader should be made aware that axis in panel B is reversed. It is very hard to see the different colours used in Figure H, in particular the lavender line is hard to read.

Reviewer #2 (Remarks to the Author):

The manuscript « Transient hydrodynamic effects influence organic carbon signatures in marine sediments » submitted by Magill et al. to Nature Communications is a novel and elegant work based on 126 radiocarbon ages on 21 paired discrete samples of clay, fine silt, coarse silt and sand compared to those obtained on bulk organic carbon and foraminifera from a southwestern Iberian margin core covering the last 25,000 years. It shows radiocarbon age offsets that are convincingly interpreted by the authors as the result of differential lateral transfer dynamics among grain-size classes during the last termination (~23,000-9,000 years ago). This differential lateral transfer would be mediated by changes in the intermediate nepheloid layer due to the dynamic of paleo-current densities. The identification of this differential lateral transfer is very important for understanding climate dynamics as it can contribute to explain the observed leads and lags among proxies preserved in different grain-size classes. This work deserves publication in Nat. Comm. after the minor revisions I have listed below:

- In lines 80-87, I have found several inconsistencies between the text, Figures 2 and 3 and Table 1 of the supplementary information. In the Table 1 and Figure 2, the maximum radiocarbon age of OC and the maximum calendar age of foraminifera are $21,409 \pm 84$ and 21,728 years, respectively while in the text the values are $20,475 \pm 775$ yr and 23,250 yr BP, respectively. In Figure 3, the oldest sample has also a calendar age of 23,250 yr BP. Similarly, the authors claim a consistent down-core difference between bulk sediment OC and foraminifera 14C ages of 1450 ± 200 years but this difference is not observed at 65-66 cm where it is only of few centuries. Finally, the radiocarbon offset between organic carbon in paired fine or coarse-silts and foraminifera is between ~1000 and 3500 years (Figure 3) and not between ~1000 and 3000 years as indicated in the text by the authors.

- Lines 97-107: In contrast with the author's statement, the relatively low radiocarbon offsets during the LGM do not transition into more moderate offsets during preliminary glacial termination. This first part of the glacial termination, ~19-17.5 kyr ago, lack of samples. This interval corresponds with a period of meltwater release into the Arctic/Nordic Seas when North Atlantic Deep Water (NADW) collapse (McManus et al., Nature, 2004; Standford et al., 2013, QSR, 2011) that may have strongly affected the northeast Atlantic margin and, consequently the 14C ages. I suggest to the authors that they rephrase this paragraph, and replace the solid lines between LGM and HS 1 with dashed lines in Figure 3A, 3B and 3F.

- Line 128: Replace "...the MOW descended to at least 2600 mbsl during past Heinrich stadials as a consequence of decreased Atlantic-Mediterranean seawater exchange" with "...the MOW ... as a consequence of increased...".

- Lines 151-152: The authors suggest that periods of strong MOW flow would result in more fine grained terrigenous clastic transport to the studied site. I wonder whether the increased aridity associated with stronger MOW (e.g. Llave et al., 2006) could enhance erosion in the adjacent landmasses and contribute to the increase arrival of fine grained terrigenous clastic particles through the canyons of Cascais, Setubal and Lisbon (Jouanneau et al., 1998, Progress in

Oceanography). The authors should discuss this issue.

- Lines 138 and 192: Replace RFS-y and RCS-y with "RFS-C and RCS-C"

- Lines 235-241: the age offsets between the different SST proxies are not clear on Figure 3H.

Also, the different blues of the SST curves are not clearly distinguishable. The authors should add a curve showing the variation of these offsets through time and chose distinct colors.

Reviewer #3 (Remarks to the Author):

Evaluation of Transient hydrodynamics effects influence organic carbon signatures in marine sediments by Magill et al.

Although I support the overall conclusion that pre-depositional processes and lateral dynamics can differentially impact proxies with different grain-size associations, I have identified multiple issues that need to be resolved before publication. The data are of high quality and the writing is clear but sometimes the authors are little too confident about their interpretation. The conclusiveness of this paper needs additional work.

Major comments:

One, I have some reservations with their neglect of bioturbation. Bioturbation can also cause size-dependent displacement and thus age offsets as the authors do acknowledge. However, they then simply state that high accumulation rates (order of 15-20 cm/kyr) are enough to ignore bioturbation. This is questionable. A simple Peclet number analysis for a mixed layer of 10 cm (maximum), a sediment accumulation of 20 cm/kyr and a biodiffusion coefficient of 2 cm²/yr, reveals that mixing is more important than sediment accumulation ($Pe < 1$). ($Pe = w * L / D = 0.02 * 10 / 2 = 0.1$). A shallower mixing layer would even increase the relative importance of bioturbation mixing vs. accumulation. Moreover, the authors do present excess 210 Pb profiles in their supplementary table. They interpret these data with an accumulation model ignoring bioturbation. This may need revision; 210 Pb vs. depth profiles are governed by radioactive decay, sediment accumulation and bioturbation (ignoring compaction). Decay is known, but accumulation and bioturbation not. You can only separate these two having another, independent tracer, like for instance the 1963 or 1986 Cs peaks. Based on their model the upper 8 cm accumulated in about 160 years, in other words, about 50 cm/kyr. Compaction can account for part but not for all the difference. Most modern sediment biogeochemists would interpret these 210Pb profiles as evidence of bioturbation. Their mixed layer is about 7 cm, not unlike the global average bioturbated layer of about 10 cm (Boudreau paper). The age of their mixed layer is about 150-200 years based on 210 Pb, but order of 1000 years based on 14C (still 4 times higher if you correct for 14C reservoir effect). This being said, bioturbators go for food (organic matter) and OM input are related to grain size sorting and lateral processing.

Two, the text presented in lines 97-107 does not align with what figure 3 shows and the authors impose an interpretation rather than letting the reader infer her/himself the story. Line: 100-101: Overall lower radiocarbon offsets... during the LGM (this is based on a single data point!) transition into more moderate offsets of 2000 plus/minus 500 yr during preliminary glacial termination (not supported by a single data point!) and then rapidly peak to values of 1500-3000 yr during HS1 (the clays offsets are the lowest in HS1 rather than highest), and so on. For instance, line 104 after an interim of low radiocarbon offset throughout B/A interstadial, 14.7-12.8 kyr ago), the relative age differences... increase again. I really do see an increase during BA and decrease during YD. This whole section does not properly link to observation in Figure 3. Why is there no zero age in figure 3 and why are two zero offsets in Figure 2 for the clays and only one in Figure 3. Aren't these the same data.

Minor points of attention.

Line 8: delivering particulate material to ocean shelves or from ocean shelves?

Line 90: large flocculates do indeed occur, but whether they survive settling to seafloor and diagenetic alteration in sediments remains to be seen. The reference cited here, as I recall, does not consider sediment aggregates, but water-column ones. Use other literature to support this claim.

Line 125: grain-size specific resuspension has also implication for how organisms mix material around.

Line 138: ... with RFS-y I presume y should be replaced with C?

Line 140: ... versus autochthonous clays.. Rewrite, this could be misread as new clay formation (the reverse weathering story).

Line 158: where is the evidence against bioturbation. I would say evidence for substantial bioturbation.

Line 172-174: the logic needs attention here. In principle OC should decrease if accumulation rates increase (simply by dilution), but usually OC and mineral delivery to sediments co-vary and OC (wt%) are rather invariant. Burial efficiencies increase of course with accumulation.

Line 185: why specifically oxic degradation? Is pre-depositional degradation not more correct?

Line 191-194: to make this more conclusive, OM quality/composition data would be needed.

Line 232-234: age offset of grain-size fractions go from zero to almost 3000, many of which 1500-2000, yet the lag (offset) in panel H, is only of the order of 200-300 years. This needs to be reconciled. Part of this may be explained by differences among sites, but how strong is your case then.

Figure 3: why is the bulk TOC content scale presented upside down (low is up).

Finally, sometimes I have the impression that the numbered references in the text and reference list are not consistent. Please check.

Jack Middelburg, 15 december 2017

Reviewer #1 (Remarks to the Author):

The authors present a new data set on ^{14}C ages of bulk organic carbon (OC) in different grain size classes of sediments from the Iberian continental margin, which are compared with ^{14}C ages of planktic foraminifera taken to reflect depositional age. They observe large and variable offsets between the OC in different grain size fraction and the depositional age and suggest sedimentology to control these, which has further implications for expected leads and lags of proxy records associated with different grain size classes. This is a very interesting study taking a novel approach, which has great potential to improve the understanding of the effects of sedimentology on proxy records.

While I agree with the general hypothesis that differences in age of different grain size fractions likely reflect differential transport histories of the respective materials in response to their hydrodynamic properties, I am not convinced by the suggested mechanisms put forward by the authors to explain their observations. First of all, I do not see the “strong positive relationship” (line 122) described for down-core values of age offsets between fine silt and foraminifera with XRF derived $\ln(\text{Zr}/\text{Al})$ values (as a side note: in the text often referred to as the linear ratio, while in the plots the logarithmic version is presented – this should be adjusted).

- There are obvious visual linear relationships apparent between calculated $R_{\text{F-FS/CS}}$ and XRF-derived $\ln(\text{Zr}/\text{Al})$ ($r^2 = 0.242$ and 0.292 , respectively [p -values < 0.05]); however, we agree that the correlation strength of this relationship is weak-to-moderate as opposed to strong. Furthermore, we also observed that the strength of this relationship is subject ‘influential points’ (SHAK06-5K sample depths of 12-23, 24-25, and 50-51 cm [equivalent to calibrated ^{14}C ages of ~670 to 3420 yr cal BP]). Thus, we have revised this section of text to more accurately present these relationships (L104–116).
- We used the natural logarithm of Zr/Al to improve data normality (i.e., minimize residuals distribution), which is necessary for meaningful interpretations of Pearson/Spearman correlation coefficients in dynamic systems¹. However, a strong positive correlation is apparent ($r^2 = 0.995$) between absolute and transformed data values since logarithmic adjustments show a stronger influence on more extreme values of the response variable. Even so, we revised the manuscript text to explain our approach and furthermore corrected the corresponding plots (c.f., Fig. 3 legend).

While there seems to be an agreement for the broad HS1 interval (with some differences in timing, which might be explained by differences in the resolution of the two data sets), I do not observe such a relationship for the YD interval. Instead, there is a strong maximum in the age offset in the BA (or a local minimum defined by just one measurement around 14.5 kyr) followed by a decline during the YD toward the early Holocene. The authors should be more careful describing such apparent agreements, in particular since their sediment core has a very high sedimentation rate and, likely, a very good age control, as it is tuned to the very well dated record MD99-2334K (another side note: a better description of the age model used and its associated uncertainties should be provided, at least in the supplement; the figures would benefit by indicating age tie points for the records of this new core SHAK06-05K).

- No problem! We revised the Results section of this manuscript to more accurately describe apparent trends and down-core correlations in an effort to resolve this important issue (c.f., L104–116 and L151–159).
- Likewise, multiple reviewers commented that further details are needed for our age-depth models, so we have added body (c.f., Methods section) and supplemental text in addition to marking pertinent tie-points in Supplementary Fig. 1 (c.f., L74 rebuttal

below, as well). This additional figure and text describes our age models, uncertainty propagation, and the alignment techniques we used to compare down-core records.

Lateral transport of the silt fractions (coarse and fine silt) is suggested to occur over long timescales (and distances) in the intermediate nepheloid layers, and current flow strength controls the amount of laterally supplied silt. This suggestion includes the inherent assumption that material transported in INLs is pre-aged. This requires INL material to be re-suspended after having been deposited elsewhere. Such re-suspended material is typically not transported very far, as these particles settle rather rapidly. I would like the authors to present evidence that pre-aged material is entrained in these INLs. Current velocities of the currents that are supposed to transport the silt fractions should also be considered in order to evaluate the likelihood of the proposed mechanism.

- A very pragmatic approach to ascertaining ‘pre-aged’ bulk OM in nepheloid layers would be a comparison of the radiocarbon ages among grain-size sediment fractions in surface sediments and the concomitant SPM, though (unfortunately) this is well beyond the scope of this study. Even so, at a minimum, the occurrence of reworked Cretaceous nannofossils along the SW Iberian margin – and at SHAK06-5K (Fig. 3C) – suggests more ancient (pre-aged) material is transported from at least hundreds of kilometres away^{2,3} because the nearest Cretaceous outcrops occur in southeast Spain⁴, although mud volcanoes of the southeast Cadiz diapiric ridge might likewise contribute some reworked Cretaceous material⁵. Concomitant Neogene nannofossils eroded from widespread Miocene outcrops along the Algarve margin insinuate more vicinal transport (10–100 km) from shoreline eroding processes, such as sea-level transgression², although inconsistent trends observed between nannofossil distribution spectra⁶ as compared to shoreline evolution⁷ argue against decisive eustatic influences on reworking patterns since at least 25 kyr ago.
- Rather, covariates of Iberian eustatic change, such as riverine suspended loads and discharge volume⁸, could drive apparent fluctuations in allochthonous sediment flux⁹. Such mechanisms square with the data shown in Fig. 3A,B in that there is a weak-to-moderate correlation apparent between radiocarbon offsets and local Iberian margin sea-level fluctuations ($r^2 = <0.098-0.476$) that features alongside much stronger correlations with inferred benthic oxygenation (Fig. 3D), MOW flow depth (Fig. 3E) and bottom-water circulation strength (Fig. 3G), each of which are subject to indirect, though co-variable, eustatic influences on water density gradients and INL formation¹⁰⁻¹³ via nonlinear internal wave dynamics^{14,15}. The mechanisms also square with studies of recent Iberian deep-sea water masses with low ratios of POC against total SPM (*t*SPM) that indicate a ‘re-suspended’ origin of this organic carbon-rich suspended fraction¹⁶. High ratios of POC to corresponding pigment levels at depths of 1000 m and deeper moreover suggest that typical POC fallout does not incorporate much fresh material, but instead features more refractory (‘pre-aged’) OM that has undergone numerous (re)suspension cycles^{16,17}.
- Interestingly, potential for sediment (re)suspension is often calculated from the ratio of force exerted by shear stress, which acts to move appertaining grains across a substrate, as compared to the effective weight of the counteracting grains (c.f., Shields mobility parameter), which is dependent upon water density gradients at the sediment interface^{14,18}. Upon resuspension, differences in settling particle velocities and their cohesiveness¹⁹ will drive size-separation of coarse material from finer sediments, which will remain in suspension comparatively longer¹⁴.
- With respect to current flow velocities, we added L254–L265. Further, we note available data indicate median grain-size of disaggregated particles within the

nepheloid fall between about 10–20 μm at depths of 2000 m and below¹⁶ along the Iberian margin. Associated benthic aggregates are in general $<125 \mu\text{m}$ in diameter²⁰. These data are also consistent with other studies suggesting progressive (re)suspension cycles within nepheloid layers results in episodic, although constant, horizontal fluxes (i.e., advection) of flocculated SPM together with rapid downslope influx of fresh phytodetritus^{16,21}. With this in mind, typical organic matter aggregates show residence times of up to several months between (re)suspension events²⁰, during which smaller aggregates ($<125 \mu\text{m}$) might travel distances of 200 km (*N.B.*, displacement distances of 10–100 km) or more with episodic²² rates of 1–10 km yr^{-1} (refs. 20,23,24).

The sediment at this core site accumulated at high but variable rates. The sedimentation rate estimates are presented in the supplement. Rough inspection reveals that the variability of the sedimentation rates resembles strongly the records of age offsets presented in Figure 3A, suggesting a coupling of sedimentation rate and the processes responsible for the age offset. Moreover, this agreement is likely better than that with $\ln(\text{Zr}/\text{Al})$ ratios, which is discussed at length in the manuscript. The implications of this observation should be discussed.

- Building upon our responses above, there is a notable correlation apparent between radiocarbon offsets and local Iberian margin sea-level fluctuations. However, there are some important differences in down-core evolution viz. slight decreases in sedimentation rate amid Heinrich Stadial 1 despite marked increases in radiocarbon offsets and the relative abundance of reworked nannofossils (c.f., Fig. 2 and Fig. 3A,C). Higher sedimentation rates amid the Last Glacial Maximum and latest Holocene also do not feature increases in reworked nannofossils. Considered together, we suggest that these data underscore a more complex or/and dynamic mechanism related to eustatic effects indirectly, such as continental hydroclimate (e.g., precipitation and fluvial discharge) in conjunction with hydrodynamic effects vis-à-vis current depth²⁵ (Fig. 3E), coastline evolution⁷ (Fig. 3B), and bottom-water oxygen conditions²⁶ (Fig. 3D).

As the core site is located at the continental margin, and the time period investigated includes a transition from low to high sea-level stand, the effect of shelf erosion induced by sea-level rise in re-distributing pre-aged sediment should be considered, too. Interestingly, the increase in age offsets (and the strongest maximum in age offsets between silt and clay) occurs approximately coevally with the onset of global sea-level rise at ~ 16.8 kyr as reconstructed by Lambeck et al., (2014). The authors should evaluate when exactly the (outer) shelf was flooded in the study area (using shelf bathymetry and the sea-level curve).

- The Lambeck et al.²⁷ sea-level curve represents global eustatic effects, which might fail to capture more local developments associated with isostatic and tectonic influences²⁸. Therefore, we adopt the more specific sea-level curve (c.f., Figure 3B) derived from estuarine sediments along the Gulf of Cadiz coast⁸ together with specific reconstructions of Iberian coastline evolution since the LGM⁷.
- There was an estimated LGM sea-level low of $\sim 130\text{m}$ in the Gulf of Cadiz^{7,29}. Amid this low-stand, fluvial channels of the Guadalquivier river incised the shelf³⁰, leading to substantial lateral erosion, especially just before the Bølling/Allerød interstadial. Thereafter, available reconstructions indicate outer shelf flooding occurred ~ 16 ky BP, followed by punctuated transgression events around 13–11 ky BP, during which the sea stood ~ 80 m below its current level⁹. With this in mind, the available sedimentological features and nanno/microfossil data suggest that fluctuations in regional ($39\text{--}43^\circ\text{N}$) lower-slope sediment flux were driven by a combination of

bioproduction, sea-level transgression, and detrital input^{7,31,32}, but deep-sea current (hydro)dynamics controlled local deposition since at least 30 kyr ago (ref. 31).

The data plots are presented without uncertainties, which may be intentional to ensure readability. However, at least for the relatively low resolution ¹⁴C data, error bars should be displayed (Fig. 3A, F).

- No problem! Age uncertainty bars have been added to corresponding panels of Figure 3, though we are interested in our peer's opinions about figure readability.

Overall, I think the study has a great potential to reveal the effect of sediment redistribution on proxy records, but more care must be taken in evaluating the possible mechanisms responsible for sediment transport, which each have a different implication.

- Thanks for these encouraging words! We agree that this study has exciting potential, and have attempted to develop this potential by refining (c.f., L171–177) and better describing possible mechanisms associated with Iberian margin sediment transport (c.f., L171–177, L193–197 and L271–275). In particular, we attempted to describe more clearly potential transmission mechanisms associated with MOW flow dynamics (i.e., advection) as compared to down-slope or vertical settling processes (c.f., Fig. 3C–E and L151–159).

Below I list a number of more specific comments:

Line 8: I think this should read "...delivering particulate material to continental slopes (or: ocean margins)..." rather than "ocean shelves", which are not considered in this study using a core retrieved from ~2.5 km depth

- We agree and have revised this sentence to read "...delivering particulate material to ocean margins...".

Line 10: "radiocarbon ...signatures measured for organic carbon in differing grain-size sediment fractions..."

- Agreed – done!

Lines 53 and following: Please provide more detail on the grain size separation methods: Did you dry sieve or wet sieve? How did you monitor recovery? Are weighted ages of OC in the grain size classes equivalent to bulk OC ages? If not, what are the potential loss mechanisms, could they be differential between compounds of different age/between different size classes?

- We have added an explicit discussion about this outstanding question, which describes our assumptions with respect to mass/organic material losses and their isotopic effects as evidenced by previous studies vis-à-vis (wet) sieving procedures.
- We also added Supplementary Fig. 2, which graphically depicts the relationship present between measured and 'mass-balanced' bulk organic signatures, which has a strong positive correlation ($r^2 = 0.718$).

Line 74: Here, more details on the age model should be provided. Reword "absolute age relationships" to "The age model", as it is not clear what the "relationship" refers to (of what with what?).

- We have added details of the age model (c.f., Methods section and Supplementary Fig. 1) and furthermore reworded this sentence to read "Age-depth models were refined...".

Line 90: I suggest to move the word "here" to after "focus"

- Agreed – this sentence is now re-worded.

Line 106: here and elsewhere (e.g., lines 122, 138): the increase in age offset, according to Figure 3, occurs prior to the YD in the BA warm period.

- These inaccuracies are now resolved to accurately present the apparent decrease in age offset that falls at the coda of Heinrich Stadial 1 and initial Bølling/Allerød, followed by pulsed increase in age offsets at Bølling/Allerød-to-Younger Dryas transition.

Line 138: unclear what the terms R_{FS-y} and R_{CS-y} refer to.

- We added the subscript ‘C’ to associated terms (i.e., R_{FS-C} and R_{CS-C}) for clarification.

Line 139: is there evidence that these fine-grained siliciclastics are indeed entrained in INLs off the Iberian margin? How far can they be transported?

- Benthic assemblages in down-core sediments suggest that most (re)suspended fine material follow discrete advective pathways³³ originated from shelf-break or upper slope (~500–1500 m depth interval)³⁴⁻³⁶ environments within the Gulf of Cadiz^{10,14,24,37} or westernmost Mediterranean³⁸. More specifically, previous studies suggest that (re)suspended fine sediments are carried by poleward MOW flow, and form a meandering permanent intermediate nepheloid layer (INL) over the mid/lower margin slope, which can detach from the seafloor and form mixed hemipelagites^{34,39}. Indeed, the existence of local INLs is inextricably linked to nepheloid layer formation within the Gulf of Cadiz and, further north, the shelf break¹⁰. As such, recent Iberian slope sedimentation is dominated by primary production in surface nepheloid layers (‘local’ input) and then rainout from intermediate nepheloid layers at depths of ~500–1500 m (‘distal’ input)³⁴.
- Other studies of the sediment dynamics along this margin demonstrate that local transmission of sediment from upper slope or shelf locations combined together with local bathymetric (morphological) features are of decisive import for sediment distribution on regional lower-slope sedimentary processes³¹, such as (re)suspension and deposition. With this in mind, during periods of MOW intensification (i.e., more vigorous and deeper current flow amid Heinrich Stadial 1), the influx of finer sediments on the southwest Portuguese margin increases consequent to more extensive wave refraction at the shelf break^{7,40} together with active channel incision and intermediate nepheloid fallout^{10,34} at the interface of MOW with fresher, cold Labrador Intermediate Water and North Atlantic Deep Water⁴¹. As such, formation of INLs is linked to bottom nepheloid layer detachment from off the shelf-break^{10,36}.

Line 148: here the authors refer to a ^{230}Th -normalized flux record, which is not presented in the figure in spite of a reference to Figure 3G.

- Thanks for catching this mistake. We have revised the sentence to read “...records of the unsupported ^{231}Pa -to- ^{230}Th ratios at SU81-18 (37.767 °N, 10.183 °W, 3135 mbsl) (Fig. 3G).”

Line 160-162: as explained above, I don’t think that it is convincingly shown that transport occurs in INLs.

- In an effort to substantiate our claims of intermediate nepheloid layer (INL) transport, we added text (c.f., L151–159 and 171–183) and Fig. 3C that links our suggested transmission mechanism with observational data and theoretic considerations; indeed, Ferreira et al.² demonstrate that high abundances of reworked Paleogene and Neogene

nannofossils (<63 μm), which are derived from mud diapirs and Guadalquivir river system, respectively⁴², occur within regional MOW flow cores as compared to coeval Atlantic water masses (e.g., surface waters and North Atlantic Deep Water).

Line 169 and 202: The term “pleniglacial” is uncommon in paleoceanography. It is more often used in terrestrial paleoenvironmental studies of the Eurasian continent, and it refers to the full glacial period prior to the LGM, which is not well resolved in the records presented here. I suggest to use more common terminology and suspect that indeed the authors meant “deglacial”

- Indeed, we meant ‘deglacial’! The incorrect term has been replaced in each instance.

Line 172: “declining percent total OC” is a bit misleading; “low” or “decreased” might be clearer.

- We revised this sentence so it now reads, “...show decreased percent total OC...”.

Line 187: Is it possible to assess/estimate how long the silt particles would reside in the INLs? Is there information on the current velocities in the respective depth?

- We have added a full paragraph (L160–183) concerning possible mechanistic relationships shared between current flow velocities, sediment (re)suspension dynamics, and differential displacement distances, which touches upon this interesting point.

Line 195-196: Here the authors mention a relationship between TOC% and OC derived from clay; it would be nice to see a figure illustrating this relationship.

- Given space constraints, we decided to add relevant data columns to Supplementary Table 1 (c.f., fraction-contribution to bulk sediment [Bulk_%] and bulk TOC_% [Bulk_{TOC%}]). Although this does not feature a distinct figure, respective data are directly amenable to regression analyses (i.e., clay fractional Bulk_{TOC%} against TOC_% [$r = 0.789$; $n = 12$, because all fractions are needed for mass balance calculations]).

Line 196 and 199: Again, the “strong parallels” between the two records are not evident from the figure; likely the authors refer mainly to the early phase of the HS1, where the age offset shows a pronounced maximum paralleled by a strong minimum in TOC%. In the YD, there is no clear correspondence; while the TOC% record has a local maximum in the BA and a minimum in the early YD which slowly increases toward the early Holocene, the age offsets are slightly higher in the BA or early YD and slowing increase through the Holocene.

- As described in an earlier response, we revised the Results section of this manuscript to more accurately describe apparent trends and down-core correlations in an effort to resolve this issue (c.f., L104–116 and L151–159).

Line 206 and following: I am confused by this paragraph, and it is not clear to me what the authors conclude from it.

- This paragraph does seem out of place, so we removed it completely!

Line 217: Is the core site really a drift deposit? Is there sedimentological evidence for this?

- Previous studies of the regional Gulf of Cadiz contourite deposition system (CDS) suggest that lower-middle slope areas of the southwest Portuguese margin are characterized by sheeted drifts alongside irregular elongated (i.e., separated/detached) drift deposits^{43,44}. Although associated CDS features are complex and their mere definition is still debated⁴⁵, the occurrence of dynamic reworked fine-grained

deposition⁴⁶ together with the consistent influence of lateral bottom currents along the slope of the southwest Portuguese margin since at least the mid-Pleistocene (c.f., MOW)⁴⁷ highlight that SHAK06-5K befits all the core characteristics of ‘mixed drifts’ that involve along-slope influences on interbedded (hemi)pelagic and down-slope (e.g., debrite) deep-water sediment facies.

- In a more subjective sense, samples are consistently poorly sorted ($\sigma_1 = 1.00\text{--}2.00$), which befits observations of typically poor sorting in clastic muddy/silty contourites, with low-intermediate degree of bioturbation, which likewise is consistent with fine-grain clastic contourites⁴⁵. Furthermore, cross-plots of the corresponding percentage coarse silt (equivalent to ‘sortable silt’ fraction)^{34,48} against the mean grain-size of the coarse silt fraction cluster around $32\pm 4\%$ and $25\pm 1\ \mu\text{m}$, respectively (c.f., Supplementary Table 1), and fall in within a range interpreted to reflect bottom-current induced (i.e., contouritic) sediment transport^{31,48}. Thus, in all likelihood, the sediments at SHAK06-5K feature a combination of deposit types, including pelagite and drifts (i.e., contourites)⁴⁵. With this in mind, (hemi)pelagic sedimentation most likely predominates associated ‘background’ deposition, with intermittent lateral bottom-current (contouritic) sedimentary processes superposed thereupon. As such, the sediments at SHAK06-5K befit the accepted definition for contourites, which is “sediments deposited or substantially reworked by the persistent action of bottom currents” (ref. 45). As defined here, contourites can also occur in conjunction (interbedded) with other sediment types, including pelagite⁴⁵.

Line 229: Does the grain size association of the biomarkers suggest more long-range transport of alkenones than of GDGTs?

- It is difficult to ascertain exact hydrodynamic consequences of differing grain-size associations since several factors might influence apparent biomarker–sediment transport histories (c.f., L254–265 and L271–278 for new, additional text).
- For instance, exact transport distances are difficult to reconstruct because available evidence indicates seafloor morphological features around Cape Sao Vicente and Coriolis effects cause dramatic current deceleration^{13,37,49} and MOW detachment² that thus curtails unobstructed hydrodynamic settling patterns of differing grain-sizes with respect to critical deposition stress^{19,48}. Transport distances (and duration) are also subject to sediment (re)suspension dynamics, which in turn are subject to changes in (dis)aggregation, particle sphericity, grain cohesiveness, riverine discharge³⁷, up-current turbidity (gravity) flow^{35,36}, bottom-current detachment³⁴, local flow velocities, and density gradients at deeper current boundaries¹⁹. Yet, despite such caveats, coarser particles (i.e., silt) fall from solution (i.e., settle out) before coeval finer material (i.e., clays) because, under similar conditions, coarser sediments are concentrated near the bottom of flows, whereas clays are distributed throughout flows, and have much higher settling velocities as compared to clays¹⁹.

Line 236: It would be helpful to specifically state which grain size class corresponds to which SST proxy.

- We have added text for clarification, which links (i) alkenones with fine silt and (ii) GDGTs with coarse silt, though there is most likely partial between sediment-fraction overlap.

Line 237 and following: If transport causes the lags in the biomarker SST records relative to the foram-derived one, a different origin/source area for the biomarkers during the periods of

stronger lateral transport would be implied, which would likely result in a shift in SST estimates (unless it is very local).

- Again, it is difficult to ascertain exact hydrodynamic consequences of differing grain-size associations since several factors might influence apparent biomarker–sediment transport histories (c.f., L254–265 and L271–278), including provenance allocation. However, we still decided to connect down-core data with modern observations (c.f., L254–265) in an effort to resolve admittedly gross constraints on sedimentary provenance.

Line 240: do you mean “in spite of rather uniform nominal grain-size distributions”?

- We agree with your suggestion, and have revised this sentence as “...despite rather uniform corresponding grain-size distributions...”.

Line 255: The YD does not show higher age offsets than the BA period! Rather, the opposite is the case.

- True! We have revised this sentence to omit the mention of the Younger Dryas and Bølling/Allerød because it doesn’t add further strength to our comparison with the mid-Holocene, which was characterized by smaller radiocarbon offsets and higher organic carbon concentrations.

Figure 1: In panel B, the estimated depth of deglacial INL is marked. It would be useful to briefly explain what this reconstruction is based on.

- We agree, so a new sentence about the model/data used for this reconstruction is now included in the legend for Fig. 1.

Figure 2: The blue shading between the lines indicating the ages of the different grain size fractions does not help much and should be removed. The age differences are better visible in Figure 3. Axis label “Sediment Fraction 14C Age” should use the unit “ky BP”, and axis label “Calendar Age” should be in “ky” (an “age” cannot be “ago”)

- No problem – we removed the shaded blue infill from this figure. Furthermore, we corrected the mislabelled figure axes as suggested to read ‘k.y. BP’.

Figure 3: In panels A, B, and F, the errors associated with the measurements should be displayed. In the caption, the reader should be made aware that axis in panel B is reversed. It is very hard to see the different colours used in Figure H, in particular the lavender line is hard to read.

- We have added propagated 1σ uncertainty bars to panels A and F. We did not include uncertainty estimates for panel B because of limited replicates or else uncertainties were within symbol bounds.
- We also mentioned the axis reversal in panel B to minimize confusion.
- We replaced the difficult-to-read lavender colour with dark green and thickened this (and the other) lines to improve readability.

Reviewer #2 (Remarks to the Author):

The manuscript « Transient hydrodynamic effects influence organic carbon signatures in marine sediments » submitted by Magill et al. to Nature Communications is a novel and elegant work based on 126 radiocarbon ages on 21 paired discrete samples of clay, fine silt, coarse silt and sand compared to those obtained on bulk organic carbon and foraminifera

from a southwestern Iberian margin core covering the last 25,000 years. It shows radiocarbon age offsets that are convincingly interpreted by the authors as the result of differential lateral transfer dynamics among grain-size classes during the last termination (~23,000-9,000 years ago). This differential lateral transfer would be mediated by changes in the intermediate nepheloid layer due to the dynamic of paleo-current densities. The identification of this differential lateral transfer is very important for understanding climate dynamics as it can contribute to explain the observed leads and lags among proxies preserved in different grain-size classes. This work deserves publication in Nat. Comm. after the minor revisions I have listed below:

In lines 80-87, I have found several inconsistencies between the text, Figures 2 and 3 and Table 1 of the supplementary information. In the Table 1 and Figure 2, the maximum radiocarbon age of OC and the maximum calendar age of foraminifera are $21,409 \pm 84$ and $21,728$ years, respectively while in the text the values are $20,475 \pm 775$ yr and $23,250$ yr BP, respectively. In Figure 3, the oldest sample has also a calendar age of $23,250$ yr BP. Similarly, the authors claim a consistent down-core difference between bulk sediment OC and foraminifera ^{14}C ages of 1450 ± 200 years but this difference is not observed at 65-66 cm where it is only of few centuries. Finally, the radiocarbon offset between organic carbon in paired fine or coarse-silts and foraminifera is between ~1000 and 3500 years (Figure 3) and not between ~1000 and 3000 years as indicated in the text by the authors.

- Thanks so much for catching this misstatement about down-core ages (*N.B.*, the incorrect value was derived from an older age model) that have now been corrected (c.f., L86–88)!
- We have also corrected the range of fraction-specific offsets to reflect the maximum age difference of ~3500 yr, and highlighted the exceptional (low) difference associated with our sample at 65–66 cm depth (c.f., L89–91).

Lines 97-107: In contrast with the author's statement, the relatively low radiocarbon offsets during the LGM do not transition into more moderate offsets during preliminary glacial termination. This first part of the glacial termination, ~19-17.5 kyr ago, lack of samples. This interval corresponds with a period of meltwater release into the Arctic/Nordic Seas when North Atlantic Deep Water (NADW) collapse (McManus et al., Nature, 2004; Stanford et al., 2013, QSR, 2011) that may have strongly affected the northeast Atlantic margin and, consequently the ^{14}C ages. I suggest to the authors that they rephrase this paragraph, and replace the solid lines between LGM and HS 1 with dashed lines in Figure 3A, 3B and 3F.

- These inaccuracies are now resolved (i.e., rephrased) to more accurately present the apparent trend in age offset that falls at the coda of Heinrich Stadial 1 and initial Bølling/Allerød (c.f., L104–116, L132–135). We have also added dashed lines connecting points that bridge between the LGM and Heinrich Stadial 1 (c.f., Fig.3 and legend).

Line 128: Replace "...the MOW descended to at least 2600 mbsl during past Heinrich stadials as a consequence of decreased Atlantic-Mediterranean seawater exchange" with "...the MOW ... as a consequence of increased...".

- We decided to remove the entire clause ("...as a consequences of decreased Atlantic-Mediterranean seawater exchange...") because it imposes a mechanistic interpretation into an otherwise descriptive sentence without due cause.

Lines 151-152: The authors suggest that periods of strong MOW flow would result in more fine grained terrigenous clastic transport to the studied site. I wonder whether the increased

aridity associated with stronger MOW (e.g. Llave et al., 2006) could enhance erosion in the adjacent landmasses and contribute to the increase arrival of fine grained terrigenous clastic particles through the canyons of Cascais, Setubal and Lisbon (Jouanneau et al., 1998, Progress in Oceanography). The authors should discuss this issue.

- It is difficult to disentangle the sources or/and the mechanisms responsible for increased terrigenous input without discrete molecular isotopic analyses of biomarkers in corresponding grain-size sediment fractions (*N.B.*, Ausin et al. *in preparation* will focus on alkenone-specific radiocarbon analyses). Even so, down-core sediment bulk C/N ratios at SHAK06-5K feature values indicate marine-dominated OM (Supplmentary Table 1). Further, studies at U1385 highlight that terrigenous material from rivers do not (directly) reach the Principe de Avis plateau^{37,50}, upon which SHAK06-5K lies. Respective studies indicate relative input from siliclastics as compared to carbonate at U1385 is influenced by glacioeustatic effects on centennial and longer timescales⁵⁰, though these effects do not appear to have a marked impact on apparent terrigenous input to southwest Portuguese margin slopes⁵¹. Rather, regional increases in rainfall lead to (i) increased fluvial discharge of freshwater (which influences MOW formation and buoyancy)^{52,53} and discharge of terrigenous clays⁵¹) and (ii) increases in chemical weathering on vegetated landscapes (which influences supplies and discharge of terrigenous clays⁵¹), with both mechanisms acting in-phase³⁷.

Lines 138 and 192: Replace RFS-y and RCS-y with “RFS-C and RCS-C“

- Both instances are now revised accordingly.

Lines 235-241: the age offsets between the different SST proxies are not clear on Figure 3H. Also, the different blues of the SST curves are not clearly distinguishable. The authors should add a curve showing the variation of these offsets through time and chose distinct colors.

- It is difficult to show centennial ¹⁴C differences among proxy records on a 25 kyr age axis. Moreover, we used Analyseries in our analyses of lead/lag phase relationships, which computes cross-covariance of down-core records and then interpolates each to create evenly spaced data at intervals commensurate with the lowest time resolution averaged throughout the entire record (*N.B.*, bin intervals of Fig. 3H are equal to about 250 yr). Thus, discrete age offsets are sometimes difficult to observe since we show original data as opposed to interpolated data, which we used for establishing phase relationships.
- Regardless, we revised the colour scheme of Fig. 3H to include more contrast (e.g., green replaced lavender) and thicker lines to improve readability.
- We also added Supplementary Fig. 3 featuring pertinent SST reconstructions derived from at MD95-2042. The records are all derived from identical down-core sample intervals, thus removing potential biases associated with absolute age differences among stratigraphically aligned (paleo)oceanographic records¹.

Reviewer #3 (Remarks to the Author):

Evaluation of Transient hydrodynamics effects influence organic carbon signatures in marine sediments by Magill et al.

Although I support the overall conclusion that pre-depositional processes and lateral dynamics can differentially impact proxies with different grain-size associations, I have identified multiple issues that need to be resolved before publication. The data are of high

quality and the writing is clear but sometimes the authors are little too confident about their interpretation. The conclusiveness of this paper needs additional work.

- We have made a major effort to temper the conclusiveness of our interpretations, and instead have attempted to more clearly describe the cantilevered logic we used to connect data – which is straightforward to discuss – with our mechanistic interpretations, which are much more subjective.

Major comments

One, I have some reservations with their neglect of bioturbation. Bioturbation can also cause size-dependent displacement and thus age offsets as the authors do acknowledge. However, they then simply state that high accumulation rates (order of 15-20 cm/kyr) are enough to ignore bioturbation. This is questionable.

- This point is well founded (to our chagrin...); thus, we added text (c.f., L126–128 and L189–193) that highlights ichnofabric evidence in analogous, vicinal down-core records at U1385 (ref. 54), M39036 (ref. 55), and PO200-10-8-2 (ref. 56) to compliment the more subjective evidence vis-à-vis sedimentation rates and ²¹⁰Pb mixed-layer depth (c.f., Supplementary Table 1). Additional details about this issue are discussed below, as well (c.f., L158 below).

A simple Peclet number analysis for a mixed layer of 10 cm (maximum), a sediment accumulation of 20 cm/kyr and a biodiffusion coefficient of 2 cm²/yr, reveals that mixing is more important than sediment accumulation ($Pe < 1$). ($Pe = w * L / D = 0.02 * 10 / 2 = 0.1$). A shallower mixing layer would even increase the relative importance of bioturbation mixing vs. accumulation. Moreover, the authors do present excess ²¹⁰Pb profiles in their supplementary table. They interpret these data with an accumulation model ignoring bioturbation. This may need revision; ²¹⁰Pb vs. depth profiles are governed by radioactive decay, sediment accumulation and bioturbation (ignoring compaction). Decay is known, but accumulation and bioturbation not. You can only separate these two having another, independent tracer, like for instance the 1963 or 1986 Cs peaks. Based on their model the upper 8 cm accumulated in about 160 years, in other words, about 50 cm/kyr. Compaction can account for part but not for all the difference. Most modern sediment biogeochemists would interpret these ²¹⁰Pb profiles as evidence of bioturbation. Their mixed layer is about 7 cm, not unlike the global average bioturbated layer of about 10 cm (Boudreau paper). The age of their mixed layer is about 150-200 years based on ²¹⁰Pb, but order of 1000 years based on ¹⁴C (still 4 times higher if you correct for ¹⁴C reservoir effect). This being said, bioturbators go for food (organic matter) and OM input are related to grain size sorting and lateral processing.

- We addressed this issue indirectly in our revisions with the addition of ichnofabric evidence (c.f., L127) and down-core reworking percentages (c.f., Fig. 3B). Below, we also include results of biodiffusion models, which can also be added as a supplementary figure if need be.

Rebuttal Fig. 1: Heat map of optimal parameter solutions for the biodiffusive model. In this heat map, warmer colors (dark red) indicate the optimal solution pairs for sedimentation rate (S , $\text{cm } 1000\text{yr}^{-1}$) and the biodiffusive coefficient (D_b , $\text{cm}^2 \text{yr}^{-1}$) given ^{210}Pb activity at depth zero ($A_0 = 44.5 \text{ dpm g}^{-1}$). The parameters were optimized by minimizing the sum of the negative log-likelihood between observed and modeled ^{210}Pb activity (A_x) for each depth (x) using the formula⁵⁷: $A_x = A_0 \text{e}^{x(S - \sqrt{S^2 + 4\lambda D_b}) / (2D_b)^{-1}}$

Two, the text presented in lines 97-107 does not align with what figure 3 shows and the authors impose an interpretation rather than letting the reader infer her/himself the story.

- Aspects of this issue were also raised by other reviewers (c.f., L97-107 rebuttal); therefore, we added text that describes – rather than interprets – the corresponding patterns shared between radiocarbon offsets and complementary proxies of (paleo)oceanographic variability. This revision is accompanied by poignant text revisions in the same paragraph to resolve inconsistencies as compared to Fig. 3.

Line: 100-101: Overall lower radiocarbon offsets... during the LGM (this is based on a single data point!) transition into more moderate offsets of 2000 plus/minus 500 yr during preliminary glacial termination (not supported by a single data point!) and then rapidly peak to values of 1500-3000 yr during HS1 (the clays offsets are the lowest in HS1 rather than highest), and so on. For instance, line 104 after an interim of low radiocarbon offset throughout B/A interstadial, 14.7-12.8 kyr ago, the relative age differences... increase again. I really do see an increase during BA and decrease during YD. This whole section does not properly link to observation in Figure 3. Why is there no zero age in figure 3 and why are two zero offsets in Figure 2 for the clays and only one in Figure 3. Aren't these the same data.

- Our original text ‘inflated’ the interpretive significance of limited LGM data, so we revised/tempered the sentence to read, ‘A lower radiocarbon offset...’, but we are also open to reviewer suggestions about how we can further improve this sentence without distracting from its descriptive clarity.
- The above revision comes in addition to more extensive changes in our manuscript text to square descriptions of data with their associated figure(s).
- With respect to data offset between sediment fractions viz. clays and foraminifera, we omitted the zero value of Fig. 3A for aesthetic reasons, which we still favour. However, we accept ^{14}C differences of <500 years are difficult to see in Fig. 2 because of the visual disparities with respect to age scales. Suggestions are welcome, and for now, we added text to Fig. 2 legend that highlights expanded data axes of Fig. 3 and the contents of Supplementary Table 1.

Minor points of attention

Line 8: delivering particulate material to ocean shelves or from ocean shelves?

- Both! We revised this sentence to read, "...interplays among processes delivering particulate material to and from ocean margins...".

Line 90: large flocculates do indeed occur, but whether they survive settling to seafloor and diagenetic alteration in sediments remains to be seen. The reference cited here, as I recall, does not consider sediment aggregates, but water-column ones. Use other literature to support this claim.

- This is an important distinction, and therefore we have augmented the references to include additional literature sources specific about benthic aggregates and boundary-layer sediment dynamics.

Line 125: grain-size specific resuspension has also implication for how organisms mix material around.

- We agree with this statement, but – in an effort to maintain manuscript focus – have decided to reword the sentence so our assertion about bottom-current influences on size-specific re-suspension is relegated to a subordinate clause as opposed to a main subject.

Line 138: ...with RFS-y I presume y should be replaced with C?

- Indeed, the 'y' terms have been replaced with 'C' both here and throughout the manuscript to improve readability.

Line 140: ... versus autochthonous clays.. Rewrite, this could be misread as new clay formation (the reverse weathering story).

- Agreed. We decided to remove corresponding parenthetical information outright to avoid future confusion.

Line 158: where is the evidence against bioturbation. I would say evidence for substantial bioturbation.

- We revised this sentence, as well as some others (c.f., L126–128 and L189–193), to more accurately present the degree (low-moderate) and tier (middle) of bioturbation in analogous down-core sediments at PO200-10-8/6/4-2 (refs. 56,58) located <10 km away from SHAK06-5K that have similar water depths (i.e., 2500±200 m).

Line 172-174: the logic needs attention here. In principle OC should decrease if accumulation rates increase (simply by dilution), but usually OC and mineral delivery to sediments co-vary and OC (wt%) are rather invariant. Burial efficiencies increase of course with accumulation.

- This is a very good observation, which we attempted to rectify by removing the contradictory perspectives inherent to our argument. That is, we removed the words 'Yet', 'despite' and 'counterintuitive', which together make our sequence of logic more...well, logical.

Line 185: why specifically oxic degradation? Is pre-depositional degradation not more correct?

- You make a convincing point! Therefore, we replaced 'oxic' with "...progressive pre-depositional degradation." We also added a more applicable reference.

Line 191-194: to make this more conclusive, OM quality/composition data would be needed.

- We agree that these data would benefit our manuscript, but – alas – we are yet to complete such analyses outside of bulk C/N ratios (c.f., Supplementary Table 1). Therefore, we added a clause about the speculative nature of our conclusions in lieu of biomarker molecular and isotopic measurements among grain-size sediment fractions.

Line 232-234: age offset of grain-size fractions go from zero to almost 3000, many of which 1500-2000, yet the lag (offset) in panel H, is only of the order of 200-300 years. This needs to be reconciled. Part of this may be explained by differences among sites, but how strong is your case then.

- The sentences in question discuss relative age offsets apparent between concurrent fine silts versus coarse silts, as opposed to ‘absolute’ age offset between sediment fractions and foraminifera. This distinction is important because the offset (lag) discussed in this paragraph (c.f., L266–278) is between silt fractions only. The offset between either silt fraction and foraminifera is somewhat larger (500–1000 yr), though still far less than the differences shown in Fig. 3A. We suggest that these discrepancies are consequent to differences in mixing proportions among proxies (i.e., relative abundance of ‘fresh’ vs. older material in corresponding proxy pools) in combination with uncertainties or natural fluctuations in sedimentary particle associations. That is, we assume specific and invariable molecular sediment-fraction associations (i.e., alkenones with fine silt; GDGTs with coarse silt) that might be over simplified for our system. Indeed, large discrepancies can arise when measured ^{14}C differences among grain-size fractions are compared to apparent lead/lag phase relationships among proxies signals derived from bulk sediments.
- Discrepancies in age offsets as compared to signal lead/lag phase relationships might likewise arise if two or more source areas share similar oceanographic conditions, such as sea-surface temperature, since apparent lead/lag patterns are related to offsets in signal timing (phase) as opposed to age *sensu stricto*.
- We hesitate to include a discussion of this issue at the moment because of its complexity, but likewise welcome reviewer opinions about the matter!

Figure 3: why is the bulk TOC content scale presented upside down (low is up).

- Bulk TOC% (panel B) is shown upside down to facilitate comparison amongst down-core records (i.e., most readers find it easier to visualize positive, as opposed to inverse, relationships). This confusing point was also noted by other reviewers; as such, we added text to Fig. 3 (legend) to underscore our reasoning.

Finally, sometimes I have the impression that the numbered references in the text and reference list are not consistent.

- Embarrassingly, you are entirely correct! We have reviewed all the references again, especially given the new references added herein.

Jack Middelburg, 15 December 2017

References

- 1 Weltje, G. J. & Tjallingii, R. Calibration of XRF core scanners for quantitative geochemical logging of sediment cores: theory and application. *Earth and Planetary Science Letters* **274**, 423-438 (2008).
- 2 Ferreira, J., Cachão, M. & González, R. Reworked calcareous nannofossils as ocean dynamic tracers: the Guadiana shelf case study (SW Iberia). *Estuar. Coast. Shelf. S.* **79**, 59-70 (2008).
- 3 Guerreiro, C., de Stigter, H., Cachão, M., Oliveira, A. & Rodrigues, A. Coccoliths from recent sediments of the central Portuguese margin: Taphonomical and ecological inferences. *Mar. Micropaleontol.* **114**, 55-68 (2015).
- 4 Flinch, J. F., Bally, A. W. & Wu, S. Emplacement of a passive-margin evaporitic allochthon in the Betic Cordillera of Spain. *Geology* **24**, 67-70 (1996).
- 5 Palomino, D. *et al.* Multidisciplinary study of mud volcanoes and diapirs and their relationship to seepages and bottom currents in the Gulf of Cádiz continental slope (northeastern sector). *Mar. Geol.* **378**, 196-212 (2016).
- 6 Incarbona, A. *et al.* Primary productivity variability on the Atlantic Iberian Margin over the last 70,000 years: evidence from coccolithophores and fossil organic compounds. *Paleoceanography* **25**, doi:10.1029/2008PA001709 (2010).
- 7 Dias, J., Boski, T., Rodrigues, A. & Magalhães, F. Coast line evolution in Portugal since the Last Glacial Maximum until present—a synthesis. *Mar. Geol.* **170**, 177-186 (2000).
- 8 Delgado, J. *et al.* Sea-level rise and anthropogenic activities recorded in the late Pleistocene/Holocene sedimentary infill of the Guadiana Estuary (SW Iberia). *Quaternary Sci. Rev.* **33**, 121-141 (2012).
- 9 Hanebuth, T. J., Zhang, W., Hofmann, A. L., Löwemark, L. A. & Schwenk, T. Oceanic density fronts steering bottom-current induced sedimentation deduced from a 50 ka contourite-drift record and numerical modeling (off NW Spain). *Quaternary Sci. Rev.* **112**, 207-225 (2015).
- 10 Oliveira, A. *et al.* Nepheloid layer dynamics in the northern Portuguese shelf. *Prog. Oceanogr.* **52**, 195-213 (2002).
- 11 Garel, E., Pinto, L., Santos, A. & Ferreira, Ó. Tidal and river discharge forcing upon water and sediment circulation at a rock-bound estuary (Guadiana estuary, Portugal). *Estuar. Coast. Shelf. S.* **84**, 269-281 (2009).
- 12 Rogerson, M., Rohling, E. J., Bigg, G. R. & Ramirez, J. Paleoceanography of the Atlantic-Mediterranean exchange: Overview and first quantitative assessment of climatic forcing. *Rev. Geophys.* **50** (2012).
- 13 Llave, E. *et al.* Bottom current processes along the Iberian continental margin. *Boletín Geológico y Minero* **126**, 219-256 (2015).
- 14 Quaresma, L. S., Vitorino, J., Oliveira, A. & da Silva, J. Evidence of sediment resuspension by nonlinear internal waves on the western Portuguese mid-shelf. *Mar. Geol.* **246**, 123-143 (2007).
- 15 Bourgault, D., Morsilli, M., Richards, C., Neumeier, U. & Kelley, D. E. Sediment resuspension and nepheloid layers induced by long internal solitary waves shoaling orthogonally on uniform slopes. *Continental Shelf Research* **72**, 21-33 (2014).
- 16 Thomsen, L. & Van Weering, T. C. Spatial and temporal variability of particulate matter in the benthic boundary layer at the NW European Continental Margin (Goban Spur). *Prog. Oceanogr.* **42**, 61-76 (1998).
- 17 Bao, R. *et al.* Widespread dispersal and aging of organic carbon in shallow marginal seas. *Geology* **44**, 791-794 (2016).

- 18 Rogerson, M., Bigg, G. R., Rohling, E. & Ramirez, J. Vertical density gradient in the
eastern North Atlantic during the last 30,000 years. *Clim. Dynam.* **39**, 589-598 (2012).
- 19 McCave, I. Size sorting during transport and deposition of fine sediments: sortable silt
and flow speed. *Developments in Sedimentology* **60**, 121-142 (2008).
- 20 Pando, S., Juliano, M., García, R., de Jesus Mendes, P. & Thomsen, L. Application of
a lagrangian transport model to organo-mineral aggregates within the Nazaré canyon.
Biogeosciences **10**, 4103 (2013).
- 21 Thomsen, L. & Gust, G. Sediment erosion thresholds and characteristics of
resuspended aggregates on the western European continental margin. *Deep-Sea Res. Pt. I* **47**, 1881-1897 (2000).
- 22 Martín, J., Palanques, A., Vitorino, J., Oliveira, A. & de Stigter, H. C. Near-bottom
particulate matter dynamics in the Nazaré submarine canyon under calm and stormy
conditions. *Deep-Sea Res. Pt. II* **58**, 2388-2400 (2011).
- 23 Davies, A. M., Xing, J., Huthnance, J. M., Hall, P. & Thomsen, L. Models of near-
bed dynamics and sediment movement at the Iberian margin. *Prog. Oceanogr.* **52**,
373-397 (2002).
- 24 Vitorino, J., Oliveira, A., Jouanneau, J. & Drago, T. Winter dynamics on the northern
Portuguese shelf. Part 1: physical processes. *Prog. Oceanogr.* **52**, 129-153 (2002).
- 25 Rogerson, M. *et al.* A dynamic explanation for the origin of the western
Mediterranean organic-rich layers. *Geochem. Geophys. Geosy.* **9** (2008).
- 26 Rodrigo-Gámiz, M., Martínez-Ruiz, F., Rampen, S., Schouten, S. & Sinninghe
Damsté, J. Sea surface temperature variations in the western Mediterranean Sea over
the last 20 kyr: A dual-organic proxy (UK' 37 and LDI) approach. *Paleoceanography*
29, 87-98 (2014).
- 27 Lambeck, K., Rouby, H., Purcell, A., Sun, Y. & Sambridge, M. Sea level and global
ice volumes from the Last Glacial Maximum to the Holocene. *P. Natl. Acad. Sci. USA*
111, 15296-15303 (2014).
- 28 Allin, J. R., Hunt, J. E., Clare, M. A. & Talling, P. J. Eustatic sea-level controls on the
flushing of a shelf-incising submarine canyon. *GSA Bulletin* **130**, 222-237 (2018).
- 29 Lobo, F., Hernández-Molina, F., Somoza, L., Del Río, V. D. & Dias, J. Stratigraphic
evidence of an upper Pleistocene TST to HST complex on the Gulf of Cádiz
continental shelf (south-west Iberian Peninsula). *Geo.-Mar. Lett.* **22**, 95-107 (2002).
- 30 Wolf, D., Seim, A. & Faust, D. Fluvial system response to external forcing and
human impact—Late Pleistocene and Holocene fluvial dynamics of the lower
Guadalete River in western Andalucía (Spain). *Boreas* **43**, 422-449 (2014).
- 31 Bender, V. B. *et al.* Control of sediment supply, palaeoceanography and morphology
on late Quaternary sediment dynamics at the Galician continental slope. *Geo.-Mar.
Lett.* **32**, 313-335 (2012).
- 32 García, M. *et al.* Erosive sub-circular depressions on the Guadalquivir Bank (Gulf of
Cadiz): Interaction between bottom current, mass-wasting and tectonic processes.
Mar. Geol. **378**, 5-19 (2016).
- 33 Bower, A. S., Serra, N. & Ambar, I. Structure of the Mediterranean Undercurrent and
Mediterranean Water spreading around the southwestern Iberian Peninsula. *Journal of
Geophysical Research: Oceans* **107** (2002).
- 34 McCave, I. & Hall, I. Turbidity of waters over the Northwest Iberian continental
margin. *Prog. Oceanogr.* **52**, 299-313 (2002).
- 35 de Stigter, H. C. *et al.* Recent sediment transport and deposition in the Nazaré
Canyon, Portuguese continental margin. *Mar. Geol.* **246**, 144-164 (2007).

- 36 de Stigter, H. C. *et al.* Recent sediment transport and deposition in the Lisbon–Setúbal and Cascais submarine canyons, Portuguese continental margin. *Deep-Sea Res. Pt. II* **58**, 2321-2344 (2011).
- 37 Ducassou, E. *et al.* Origin of the large Pliocene and Pleistocene debris flows on the Algarve margin. *Mar. Geol.* **377**, 58-76 (2016).
- 38 Cacho, I., Grimalt, J. O., Sierro, F. J., Shackleton, N. & Canals, M. Evidence for enhanced Mediterranean thermohaline circulation during rapid climatic coolings. *Earth Planet. Sc. Lett.* **183**, 417-429 (2000).
- 39 Grousset, F. *et al.* Mediterranean outflow through the Strait of Gibraltar since 18,000 years BP: mineralogical and geochemical arguments. *Geo.-Mar. Lett.* **8**, 25-34 (1988).
- 40 Cascalho, J., Magalhaes, F., Dias, J. & Carvalho, A. Sedimentary unconsolidated cover of the Alentejo continental shelf (first results). *Gaia* **8**, 113-118 (1994).
- 41 Hall, I. R. & McCave, I. N. Palaeocurrent reconstruction, sediment and thorium focussing on the Iberian margin over the last 140 ka. *Earth Planet. Sc. Lett.* **178**, 151-164 (2000).
- 42 López-Galindo, A., Rodero, J. & Maldonado, A. Surface facies and sediment dispersal patterns: southeastern Gulf of Cadiz, Spanish continental margin. *Mar. Geol.* **155**, 83-98 (1999).
- 43 Pereira, R., Alves, T. M. & Cartwright, J. Post-rift compression on the SW Iberian margin (eastern North Atlantic): a case for prolonged inversion in the ocean–continent transition zone. *Journal of the Geological Society* **168**, 1249-1263 (2011).
- 44 Hernández-Molina, F. J. *et al.* Along-slope oceanographic processes and sedimentary products around the Iberian margin. *Geo.-Mar. Lett.* **31**, 315-341 (2011).
- 45 Rebesco, M., Hernández-Molina, F. J., Van Rooij, D. & Wåhlin, A. Contourites and associated sediments controlled by deep-water circulation processes: state-of-the-art and future considerations. *Mar. Geol.* **352**, 111-154 (2014).
- 46 Stow, D., Hunter, S., Wilkinson, D. & Hernández-Molina, F. The nature of contourite deposition. *Developments in sedimentology* **60**, 143-156 (2008).
- 47 Hernández-Molina, F. *et al.* Oceanographic processes and products around the Iberian margin: a new multidisciplinary approach. *Bol. Geol. Min* **126**, 279-326 (2015).
- 48 McCave, I. N. & Hall, I. R. Size sorting in marine muds: processes, pitfalls, and prospects for paleoflow-speed proxies. *Geochem. Geophys. Geosy.* **7** (2006).
- 49 Hernández-Molina, F. J. *et al.* Contourite processes associated with the Mediterranean Outflow Water after its exit from the Strait of Gibraltar: Global and conceptual implications. *Geology* **42**, 227-230 (2014).
- 50 Lofi, J. *et al.* Quaternary chronostratigraphic framework and sedimentary processes for the Gulf of Cadiz and Portuguese Contourite Depositional Systems derived from Natural Gamma Ray records. *Mar. Geol.* **377**, 40-57 (2016).
- 51 Sierro, F. J., Ledesma, S., Flores, J.-A., Torrecusa, S. & del Olmo, W. M. Sonic and gamma-ray astrochronology: cycle to cycle calibration of Atlantic climatic records to Mediterranean sapropels and astronomical oscillations. *Geology* **28**, 695-698 (2000).
- 52 Bahr, A. *et al.* Persistent monsoonal forcing of Mediterranean Outflow Water dynamics during the late Pleistocene. *Geology* **43**, 951-954 (2015).
- 53 Jiménez-Espejo, F. J. *et al.* Geochemical evidence for intermediate water circulation in the westernmost Mediterranean over the last 20kyrBP and its impact on the Mediterranean Outflow. *Global Planet. Change* **135**, 38-46 (2015).
- 54 Dorador, J. & Rodríguez-Tovar, F. J. Stratigraphic variation in ichnofabrics at the “Shackleton Site”(IODP Site U1385) on the Iberian Margin: Paleoenvironmental implications. *Mar. Geol.* **377**, 118-126 (2016).

- 55 Löwemark, L., Schönfeld, J., Werner, F. & Schäfer, P. Trace fossils as a paleoceanographic tool: evidence from Late Quaternary sediments of the southwestern Iberian margin. *Mar. Geol.* **204**, 27-41 (2004).
- 56 Baas, J. H., Mienert, J., Abrantes, F. & Prins, M. A. Late Quaternary sedimentation on the Portuguese continental margin: climate-related processes and products. *Palaeogeogr. Palaeocl.* **130**, 1-23 (1997).
- 57 Nittrouer, C., DeMaster, D., McKee, B., Cutshall, N. & Larsen, I. The effect of sediment mixing on Pb-210 accumulation rates for the Washington continental shelf. *Mar. Geol.* **54**, 201-221 (1984).
- 58 Baas, J., Schönfeld, J. & Zahn, R. Mid-depth oxygen drawdown during Heinrich events: evidence from benthic foraminiferal community structure, trace-fossil tiering, and benthic $\delta^{13}\text{C}$ at the Portuguese Margin. *Mar. Geol.* **152**, 25-55 (1998).

Reviewers' comments:

Reviewer #1 (Remarks to the Author):

General note: The line numbers given in the rebuttal do not agree with the line numbers in the revised manuscript, which makes it harder to assess whether appropriate changes were made.

Overall, the authors have responded in detail to each of the points raised in my review of the originally submitted manuscript. However, the revised manuscript still stresses the importance of intermediate nepheloid layers (as opposed to bottom nepheloid layers, which could also be created during sea-level rise, or enhanced supply of riverine material) for lateral particle transport. I think this is not substantiated by the data, as the paper does not provide evidence that pre-aged organic matter is indeed entrained in these INLs. The text should be more thoroughly revised to make clear that transport in INLs is one of several possible scenarios.

Replies to my comments on the original manuscript:

-The authors argue that the putative "strong positive relationship between downcore values of age offsets with XRF-derived $\ln(\text{Zr}/\text{Al})$ values is apparent visually and from linear regression; it would be good if these regressions were shown in the SI. The r^2 values of 0.242 and 0.292 do not convince this reviewer even of a moderate positive relationship

- age model: a new paragraph was added describing how the age model was constructed. I am wondering why the authors used the calibration software Calib in the slightly outdated version 6.0?

- description of the temporal evolution of age offsets (re. Fig. 3f). I am still not satisfied with the way the age offsets are described. For instance, the authors state: "low-to-intermediate average radiocarbon offsets ...persist through the mid-Holocene", while no mention is made of the rather large age offsets of up to 2000 yr observed in the late Holocene. These offsets are in fact much higher than the putative "peak" during the "B/A-YD transition". While I agree that local maxima may be apparent, in particular during the early HS1, the data from the early BA on to my eyes only show a gradual increase; there are two contradicting local maxima for the two FB and CS grain size fractions.

- In my previous comment I suggested that lateral transport of pre-aged material might not happen in the intermediate nepheloid layer but instead by other ways. The reply presented by the author does not address this concern but instead lists numerous lines of evidence for the occurrence of lateral transport, which is not doubted. I would like to refer to a study on intermediate and bottom nepheloid layer dynamics, which shows that most of the material in INTERMEDIATE nepheloid layers is actually very fresh (Karakas et al, 2006, JGR). The references cited in the rebuttal refer to BOTTOM nepheloid layers, which have inherently different erosion and re-deposition dynamics. I would like to suggest that the author make a clear distinction between intermediate and bottom nepheloid layers and refrain from directly attributing transport to INLs (e.g., line 168). I also note that in the rebuttal, several months are cited as average residence times in re-suspension events of organic matter aggregates. This estimated timescale for lateral transport is in conflict with the data presented here, where organic matter in the fine fraction is up to 3500 yr older than the depositional age.

- I am not entirely satisfied with the reply to my comment suggesting a potential tie between sedimentation rate and age offset. The authors should present a plot of sedimentation rate along with the other data. In line 195, they refer to figure 3 a,b in context of a correlation between age offsets and sedimentation rates, but on the figure, plots of age offsets (a) and sea-level and bulk TOC% (b) are shown.

Likewise, I acknowledge the inclusion of the local sea-level reconstruction. However, the first

derivative of this record, i.e., the rate of sea-level change would be more appropriate. From visual inspection of the local sea-level curve it appears that the strongest increase in sea-level happened in the second half of the younger Dryas right after 12 kyr BP, probably in line with the global sea-level rise.

When inspecting the references cited for this record, I could not find the data to support the displayed curve (ref 79 provides a plot of a previously published sea-level reconstruction with some relatively strong oscillations, while ref 80 covers only the time period from about 12 kyr to present). Moreover, it seems from the plot displayed in Figure 3b that for the intervals between ~23 kyr BP and 17.5 kyr BP and between 17.5 kyr BP and ~13 kyr BP, no local data exist and the record displayed is a linear interpolation between existing data points. If this is the case (displaying fixed data points might have helped in evaluating this), the global reconstruction might be more suitable for assessing this question.

A one-to-one relationship between sedimentation rate and age offsets would be rather surprising, as both depend on a variety of different factors, as rightly stated in the reply letter. However, the multiple potential influences on the age offsets and the different mechanism of sediment remobilization should be discussed in more detail in the manuscript. As is, the discussion remains centered around the assumed intermediate nepheloid layer transport, which I doubt is the proper mechanism.

-Figure 3, label of x-axis: Still the misleading unit (k.y. ago) is displayed. Please change to ky BP

Further notes:

Line 108-109: Here, it sounds like the decrease in age offsets is due to an unexpectedly low TOC age, while from figure 2 it looks like the decrease is due to a older than expected foram age.

Line 175: It is not clear to me what is meant here. Do the authors imply that redox conditions impact the age of OM in sediment and SPM? How would this occur?

Line 195 and following: Here a reference is made to Fig 3a,b when discussion down-core sedimentation rate. However, sedimentation rate is not displayed in this figure. Please add a plot of sedimentation rate (see comment above). Further, the sentence starting in 196 seems grammatically incorrect. "Aspiration" seems an odd term in context of ocean circulation.

Reviewer #2 (Remarks to the Author):

I have checked the revised version of the Magill et al.'s manuscript. I feel that the points rose by the three reviewers, and particularly my own concerns, have been satisfactorily addressed. In my opinion, this manuscript is ready for publication after the following three minor modifications:

- Supplementary Figure 1: There is an inconsistency between the figure (y-axis) and the caption concerning the name of one of the sedimentary sequences. The authors should replace "MD99-2444" in the y-axis with "MD01-2444".
- Figure 3: I am concerned by the name given to the interval 17.5-14.7 ka. This interval corresponds with the Heinrich event 1, i.e. the massive iceberg discharges of the North American ice caps during the last deglaciation, and not to the HS 1 that lasted longer (19/18 – 14.7 ka) because it includes the previous European iceberg discharges (Standford et al., 2011, QSR). The authors should replace "HS 1" with "HE 1" or enlarge the yellowish band.
- Line 175: In the main text the authors should introduce the meaning of SPM (Suspended Particular Matter).

Reviewer #3 (Remarks to the Author):

I have read the rebuttal of Magill et al. and believe it is close to ready for publication. The authors have done an excellent job in revising the manuscript and addressing most comments.

I recommend that they include the rebuttal Fig. 1 (about accumulation vs. bioturbation) in the SI. Why? I do not consider the ichnofabric argument against bioturbation as strong evidence. And this rebuttal Fig. 1 shows precisely why bioturbation cannot be ignored (as also evidenced by the lack of Cs peak (xls file0, but smearing all over, given the resolution; and a Pe of 1 based on their data, implying mixing and sediment accretion are similarly important). Also, a lack of a mixed layer (with ^{210}Pb excess, L147) does not mean there is no bioturbation. A mixed layer is usually only visible for longer-lived radionuclides. Is the Zonneveld paper the best reference for this?

L131: is it wise to use the word 'coda' for an international audience dominated by non-native speakers nowadays?

L174: .. that, in turn, impact... (this logic requires a reference).

Jack Middelburg, May 7, 2018

Reviewers' comments:

Reviewer #1 (Remarks to the Author):

General note: The line numbers given in the rebuttal do not agree with the line numbers in the revised manuscript, which makes it harder to assess whether appropriate changes were made.

- We have attempted to resolve this issue, as it seems to arise as a consequence of line-number differences either (1) between manuscript drafts (i.e., original vs. revised) or/and (2) from changes when switching between 'Track Changes' options (e.g., simple- vs. no-markup). Regardless, we have double-checked line callouts in our responses in hopes of making this manuscript less difficult to review!

Overall, the authors have responded in detail to each of the points raised in my review of the originally submitted manuscript. However, the revised manuscript still stresses the importance of intermediate nepheloid layers (as opposed to bottom nepheloid layers, which could also be created during sea-level rise, or enhanced supply of riverine material) for lateral particle transport. I think this is not substantiated by the data, as the paper does not provide evidence that pre-aged organic matter is indeed entrained in these INLs. The text should be more thoroughly revised to make clear that transport in INLs is one of several possible scenarios.

- Agreed; if the text is unclear to our reviewers, then we need to revise accordingly, given – in our experience – if it is unclear to our reviewer(s), then it will be unclear to our readers! With this in mind, we have made a focused attempt to clarify problematic discussions in our manuscript (e.g., highlight that intermediate nepheloid layers are one of several likely/potential drivers of differential down-core organic radiocarbon offsets [c.f., the more comprehensive responses detailed below]).

Replies to my comments on the original manuscript: The authors argue that the putative “strong positive relationship between downcore values of age offsets with XRF-derived $\ln(\text{Zr}/\text{Al})$ values is apparent visually and from linear regression; it would be good if these regressions were shown in the SI. The r^2 values of 0.242 and 0.292 do not convince this reviewer even of a moderate positive relationship

- We appreciate our reviewer's candor, and have added the suggested figures showing plots of down-core offsets against $\ln(\text{Zr}/\text{Al})$ below. However, we hesitate to add the same figures to our Supplementary Appendix since the same information is straightforward to recreate from Supplementary Table 1.

Age model: a new paragraph was added describing how the age model was constructed. I am wondering why the authors used the calibration software Calib in the slightly outdated version 6.0?

- There is no specific reason for our use of the older version outside of historical legacy... As such, we revised the age model to use a more recent version of Calib (7.1 [c.f., ref. 33]), though there is no significant impact to our age models.

Description of the temporal evolution of age offsets (re. Fig. 3f). I am still not satisfied with the way the age offsets are described. For instance, the authors state: “low-to-intermediate average radiocarbon offsets ...persist through the mid-Holocene”, while no mention is made of the rather large age offsets of up to 2000 yr observed in the late Holocene. These offsets are in fact much higher than the putative “peak” during the “B/A-YD transition”. While I agree that local maxima may be apparent, in particular during the early HS1, the data from the early BA on to my eyes only show a gradual increase; there are two contradicting local maxima for the two FB and CS grain size fractions.

- We have addressed this interesting point in a new sentence at the end of the results section (L119-121) that describes an increase in average radiocarbon offsets amid the last few millennia, and furthermore underscore its interpretational link to anthropogenic activities.

In my previous comment I suggested that lateral transport of pre-aged material might not happen in the intermediate nepheloid layer but instead by other ways. The reply presented by the author does not address this concern but instead lists numerous lines of evidence for the occurrence of lateral transport, which is not doubted. I would like to refer to a study on intermediate and bottom nepheloid layer dynamics, which shows that most of the material in INTERMEDIATE nepheloid layers is actually very fresh (Karakas et al, 2006, JGR). The references cited in the rebuttal refer to BOTTOM nepheloid layers, which have inherently different erosion and re-deposition dynamics. I would like to suggest that the author make a clear distinction between intermediate and bottom nepheloid layers and refrain from directly attributing transport to INLs (e.g., line 168). I also note that in the rebuttal, several months are cited as average residence times in re-suspension events of organic matter aggregates. This estimated timescale for lateral transport is in conflict with the data presented here, where organic matter in fine fraction is up to 3500 yr older than the depositional age.

- To our chagrin, this is a valid concern in light of limited data with respect to organic matter composition and age in nepheloid layers. Therefore, we now introduce ‘bottom and intermediate nepheloid layers’ together (L128). Furthermore, we replaced interpretational discussion/attribution of intermediate nepheloid layers (INLs) with the more universal term ‘nepheloid layer[s]’ (c.f., L156, L166, L170, L176, L246, L304).
- We added the aforementioned Karakas et al. (2006 [ref. 44]) to our references (c.f., L128).
- The apparent time discrepancies observed between average residence (re-suspension) time and the age-offset between differing grain-size sediment fractions (i.e., months vs. millennia) highlights an interesting point for discussion – namely, grain-size specific aging patterns are more sensitive to differential losses of fresh organic matter (i.e., relative increase in old vs. fresh carbon) than entrainment time *sensu stricto*. Indeed, there is a cumulative aging effect that develops consequent to relative changes in organic matter mixing proportions, which is superposed upon (1) differential

transmission among grain-size sediment fractions (c.f., L227-229), and (2) differential inputs among grain-size sediment fractions (c.f., Mulder et al. 2013 [ref. 13]). As such, the radiocarbon age of different bulk sediment fractions more-so represents mixing proportions with respect to differential loss (gain) of fresh (old) organic matter amid lateral transmission (c.f., Bao et al. 2016 [*Geology*]), whereas average residence in (re)suspension constrains sedimentary provenance vis-à-vis average current flow velocities, current thickness, and down-column sinking particle rates (c.f., Hanebuth et al. 2015 [ref. 56]). With the above in mind, we added text to underscore this important distinction (c.f., L260-271).

I am not entirely satisfied with the reply to my comment suggesting a potential tie between sedimentation rate and age offset. The authors should present a plot of sedimentation rate along with the other data. In line 195, they refer to figure 3 a,b in context of a correlation between age offsets and sedimentation rates, but on the figure, plots of age offsets (a) and sea-level and bulk TOC% (b) are shown.. Likewise, I acknowledge the inclusion of the local sea-level reconstruction.

- We agree with the suggestion that there should – at a minimum – be a clear graphical depiction of trends in sedimentation rate at SHAK06-5K; therefore, we amended Fig. 3 to include sedimentation rate (panel B). Hopefully, this addition makes between-proxy comparisons clearer and more straightforward.

However, the first derivative of this record, i.e., the rate of sea-level change would be more appropriate. From visual inspection of the local sea-level curve it appears that the strongest increase in sea-level happened in the second half of the younger Dryas right after 12 kyr BP, probably in line with the global sea-level rise. When inspecting the references cited for this record, I could not find the data to support the displayed curve (ref 79 provides a plot of a previously published sea-level reconstruction with some relatively strong oscillations, while ref 80 covers only the time period from about 12 kyr to present). Moreover, it seems from the plot displayed in Figure 3b that for the intervals between ~23 kyr BP and 17.5 kyr BP and between 17.5 kyr BP and ~13 kyr BP, no local data exist and the record displayed is a linear interpolation between existing data points. If this is the case (displaying fixed data points might have helped in evaluating this), the global reconstruction might be more suitable for assessing this question.

- We have added a figure panel (Fig. 3C) that displays the *rate* of sea-level change as modelled from coral data in nested Monte Carlo experiments (c.f., Stanford et al. 2011 [ref. 86]).
- This reviewer raises an interesting point that at least I (C.R.M.) had not thought of before with respect to data continuity, particularly through HE1. Therefore, we have added data derived from complementary global-scale relative sea-level reconstructions (i.e., Barbados, Bonaparte Gulf, and New Guinea) to contextualize apparent local-to-regional trends (c.f., Fig. 3C).

A one-to-one relationship between sedimentation rate and age offsets would be rather surprising, as both depend on a variety of different factors, as rightly stated in the reply letter. However, the multiple potential influences on the age offsets and the different mechanism of sediment remobilization should be discussed in more detail in the manuscript. As is, the

discussion remains centered around the assumed intermediate nepheloid layer transport, which I doubt is the proper mechanism.

- Although our discussion still focuses on nepheloid layers, we appreciate the importance of multiple drivers with respect to sediment (re)mobilization, and have ventured to (1) temper our implied tenet that intermediate nepheloid layers are a dominant driver of hydrodynamic sorting processes at SHAK06-5K (i.e., we now use the more universal term ‘nepheloid layers’), and (2) highlight the importance of less-discussed hydrodynamic drivers – such as sedimentation rate (Fig. 3B), sea-level fluctuations (Fig. 3C), and bioturbation (Supplementary Figure 4) – through figures and topical literature references (see refs. 44, 49, 50, 57, 58, 86) .

Figure 3, x-axis: Still the misleading unit (k.y. ago) is displayed. Please change to ky BP

- Thanks for catching this...again! We have definitely revised it this time.

Further notes:

Line 108-109: Here, it sounds like the decrease in age offsets is due to an unexpectedly low TOC age, but from figure 2 it looks like the decrease is due to older than expected foram age.

- Taken care of! This sentence is now re-written to more clearly point this observation out (c.f., L112-114)!

Line 175: It is not clear to me what is meant here. Do the authors imply that redox conditions impact the age of OM in sediment and SPM? How would this occur?

- The reviewer’s intuition is correct! We do intend to implicate redox conditions as an influence on organic matter age in sediment, but this implication must have come across as unclear as a couple of the reviewers raised the same issue (c.f., Reviewer #3 below). Therefore, we have added details about the effects of redox conditions on organic matter degradation in marine environments and the reference of Blair and Aller (2012 [ref. 57]) and Hedges et al. (1999 [ref. 58]) for additional background to our readers. In short, the cumulative effects of oxygen (re)exposure cause selective degradation of fresh (‘labile’) organic matter, which leaves older, more recalcitrant OM behind.

Line 195 and following: Here a reference is made to Fig 3a,b when discussion down-core sedimentation rate. However, sedimentation rate is not displayed in this figure. Please add a plot of sedimentation rate (see comment above). Further, the sentence starting in 196 seems grammatically incorrect. “Aspiration” seems an odd term in context of ocean circulation.

- As suggested, there is now a plot of sedimentation rate added to Fig. 3 (panel B).
- I (C.R.M.) didn’t find the suspected grammar mistake in the last paragraph of the ‘Potential driver(s)...’ (L190-203) section or the first paragraph of the ‘(Paleo)oceanographic implication’ (L207-222) section, but I am happy to correct the mistake if it is explained to me :)
- We agree – ‘aspiration’ is not ideal here... Therefore, we have replaced it with ‘injection’ (c.f., L184), though we are also amenable to other suggestions (e.g., ‘introduction’).

Reviewer #2 (Remarks to the Author):

I have checked the revised version of the Magill et al.'s manuscript. I feel that the points raised by the three reviewers, and particularly my own concerns, have been satisfactorily addressed. In my opinion, this manuscript is ready for publication after the following three minor modifications:

Supplementary Figure 1: There is an inconsistency between the figure (y-axis) and the caption concerning the name of one of the sedimentary sequences. The authors should replace "MD99-2444" in the y-axis with "MD01-2444".

- Another excellent find that has been revised accordingly in Supplementary Figure 1!

Figure 3: I am concerned by the name given to the interval 17.5-14.7 ka. This interval corresponds with the Heinrich event 1, i.e. the massive iceberg discharges of the North American ice caps during the last deglaciation, and not to the HS 1 that lasted longer (19/18 – 14.7 ka) because it includes the previous European iceberg discharges (Standford et al., 2011, QSR). The authors should replace "HS 1" with "HE 1" or enlarge the yellowish band.

- No problem! We decided to replace 'HS1' with 'Heinrich Event 1' (HE1) throughout the manuscript because we also mention a specific interval (c.f., L112) that likewise might lead to confusion about the age and duration of the ice-discharge event that we want to underscore (i.e., HE1).

Line 175: In the main text the authors should introduce the meaning of SPM (Suspended Particular Matter).

- Taken care of! Thanks for catching this oversight (*N.B.*, I despise when authors do not define acronyms...).

Reviewer #3 (Remarks to the Author):

I have read the rebuttal of Magill et al. and believe it is close to ready for publication. The authors have done an excellent job in revising the manuscript and addressing most comments.

I recommend that they include the rebuttal Fig. 1 (about accumulation vs. bioturbation) in the SI. Why? I do not consider the ichnofabric argument against bioturbation as strong evidence. And this rebuttal Fig. 1 shows precisely why bioturbation cannot be ignored (as also evidenced by the lack of Cs peak (xls file0, but smearing all over, given the resolution; and a Pe of 1 based on their data, implying mixing and sediment accretion are similarly important). Also, a lack of a mixed layer (with 210Pb excess, L147) does not mean there is no bioturbation. A mixed layer is usually only visible for longer-lived radionuclides. Is the Zonneveld paper the best reference for this?

- We agree with both points raised here and have revised accordingly. That is, we have included our modelled bioturbation heat-map in our supplemental file as Supplementary Figure 4.
- Likewise, we have updated the references (c.f., L135) vis-à-vis mixed layers and bioturbation to more appropriate articles viz. Nittrouer et al. (1984 [ref. 49]) and Meysman et al. (2003 [ref. 50]).

L131: is it wise to use the word ‘coda’ for an international audience dominated by non-native speakers nowadays?

- Probably not... Therefore, we changed this word to a more common (and less musical) term viz. ‘conclusion’!

L174: ...that, in turn, impact... (this logic requires a reference).

- Agreed! Therefore, we have added the reference of Blair and Aller (2012 [ref. 57]) that describes organic matter degradation in marine environments.

REVIEWERS' COMMENTS:

Reviewer #1 (Remarks to the Author):

The authors were careful to respond to all issues raised in my previous review, and I am happy with most of the responses. Some details remain, and I list them below.

Again, in spite of better efforts on the authors' side, there seems to be mismatch between line numbers in the merged pdf and those cited in the rebuttal. For instance, in the rebuttal the author claim to have changed the sentence in lines 112-114, but this sentence has not been changed compared to the previous version. Besides, changes are not marked in the pdf of R2, which makes it hard to see where changes have been made.

I appreciate that the authors produced x-y plots illustrating the relationship between age offsets and $\ln(\text{Zr}/\text{Al})$ for the rebuttal. Having seen those, my evaluation remains the same, i.e., that the relationship is rather weak if existent at all. I still suggest to tone down the statements made regarding this relationship (lines 170-177).

I apparently missed this in the previous version – but I do not find a discussion on why GDGTs are associated with coarse silt, while alkenones are entrained in the fine silt fraction (line 304)? I think this requires explanation.

Details:

Line 87: Radiocarbon is not a radiogenic isotope, as it is not produced by radioactive decay.

Line 131: This should be “particularly ‘old’ foraminifera calendar ages”

Line 130-132: The sentence “though the magnitude of these offsets appear influenced by particular ‘old’ foraminifera calendar ages as opposed to unexpectedly young gross sediment TOC.” appears to be placed in the wrong position. In my comments re: R1 I noted that the statement “Associated bulk sediment OC 14C feature a consistent down-core difference against foraminifera radiocarbon ages of 1450 ± 200 yr (Fig. 2) with one exception at 65–66 cm (~600 yr offset [Supplementary Table 1]) that fell during prominent Mediterranean sapropel 1 formation” (lines 106-109) should be complemented by pointing out that the “exception” is caused mainly by an old foram age.

Line 202: “This combination of (hydro)physical drivers is would drive...” delete “is” (this is mistake that was referred to in previous review, then line 196 and following, but was not found by the author)

Line 228: Refer here to Fig 3d (increases in reworked nannofossil flux)

Line 251 RFs-c and RCS-S trends are mentioned – refer to Fig 3G

REVIEWERS' COMMENTS:

Reviewer #1 (Remarks to the Author):

The authors were careful to respond to all issues raised in my previous review, and I am happy with most of the responses. Some details remain, and I list them below

Again, in spite of better efforts on the authors' side, there seems to be mismatch between line numbers in the merged pdf and those cited in the rebuttal. For instance, in the rebuttal the author claim to have changed the sentence in lines 112-114, but this sentence has not been changed compared to the previous version. Besides, changes are not marked in the pdf of R2, which makes it hard to see where changes have been made.

CM3: I have no excuse for this inconvenience, and have attempted (again) to match line-numbers shown in our draft (*N.B.*, the 'No Markup' numbers are shown next to our reviewer responses now). Furthermore, revisions are also available in 'Track Changes' mode within our *.docx manuscript file.

I appreciate that the authors produced x-y plots illustrating the relationship between age offsets and $\ln(Zr/Al)$ for the rebuttal. Having seen those, my evaluation remains the same, i.e., that the relationship is rather weak if existent at all. I still suggest to tone down the statements made regarding this relationship (lines 170-177).

CM3 [L115, 119]: I have attempted to tone down the emphasis on aforementioned XRF trends with respect to radiocarbon offsets by replacing phrase 'much higher' (L115) with 'stronger', and by replacing phrase 'strong parallels' with 'significant low-to-moderate strength relationships' (L119).

I apparently missed this in the previous version – but I do not find a discussion on why GDGTs are associated with coarse silt, while alkenones are entrained in the fine silt fraction (line 304)? I think this requires explanation.

CM3 [L235-238]: The mechanism(s) underlying grain-size–molecular associations remain speculative (i.e., observational), but there are some relevant insights. For instance, alkenones might be associated with the clay-sized fraction because coccoliths, which are derived from alkenone-synthesizing coccolithophores, are typically $<6 \mu\text{m}$ in size (Pedrosa-Pàimes et al. 2015). Alternatively, generic alkenone-sediment fraction associations could be related to mineral-face surface sorption on clays (Quirós-Collazos et al. 2017). In a similar sense, coarse silt-associated GDGTs might be consequent to either increased terrestrial inputs of the coarser silts (i.e., rivers or sea ice), which host GDGT-synthesizing soil bacteria, or serendipitous association between silt-rich sediments and in situ marine communities of *Thaumarchaeota* (Park et al. 2014). Given such mechanistic uncertainties and tangential importance of determining grain-

size–molecular sorption dynamics with respect to our manuscript, I hesitate to add further explanation/discussion about this otherwise interesting point!

Details:

Line 87: Radiocarbon is not a radiogenic isotope, as it is not produced by radioactive decay.

CM3 [L293]: This is an embarrassing mistake that has been corrected to read “Radiocactive carbon isotope compositions...”.

Line 131: This should be “particularly ‘old’ foraminifera calendar ages”

CM3 [L55]: Taken care of! We now use a grammatically-correct adverb, though the phrase under consideration has been moved to L53-56.

Line 130-132: The sentence “though the magnitude of these offsets appear influenced by particular ‘old’ foraminifera calendar ages as opposed to unexpectedly young gross sediment TOC.” appears to be placed in the wrong position. In my comments re: R1 I noted that the statement “Associated bulk sediment OC 14C feature a consistent down-core difference against foraminifera radiocarbon ages of 1450 ± 200 yr (Fig. 2) with one exception at 65–66 cm (~600 yr offset [Supplementary Table 1]) that fell during prominent Mediterranean sapropel 1 formation” (lines 106-109) should be complemented by pointing out that the “exception” is caused mainly by an old foram age.

CM3 [L53-56]: This is an interesting point that I fully agree with! Therefore, the explanatory phrase “though the magnitude...sediment TOC” has been moved to the first paragraph of our ‘Results and Discussion’ section.

Line 202: “This combination of (hydro)physical drivers is would drive...” delete “is” (this is mistake that was referred to in previous review, then line 196 and following, but was not found by the author)

CM3 [L147]: Thanks for catching this – again! Third time is a charm... Regardless, we have removed the word ‘is.’

Line 228: Refer here to Fig 3d (increases in reworked nannofossil flux)

CM3 [L172]: No problem – a figure call-out has been added here.

Line 251 RfS-c and RCS-S trends are mentioned – refer to Fig 3G

CM3 [L195]: Again, no problem and figure call-out has been added!